# Probing aggrephagy using chemically-induced protein aggregates

Anne F.J. Janssen[1], Eugene A. Katrukha [1], Wendy van Straaten[1], Pauline Verlhac [2], Fulvio Reggiori [2] & Lukas C. Kapitein[1]

Selective types of autophagy mediate the clearance of specific cellular components and are essential to maintain cellular homeostasis. However, tools to directly induce and monitor such pathways are limited. Here we introduce the PIM (particles induced by multimerization) assay as a tool for the study of aggrephagy, the autophagic clearance of aggregates. The assay uses an inducible multimerization module to assemble protein clusters, which upon induction recruit ubiquitin, p62, and LC3 before being delivered to lysosomes. Moreover, use of a dual fluorescent tag allows for the direct observation of cluster delivery to the lysosome. Using flow cytometry and fluorescence microscopy, we show that delivery to the lysosome is partially dependent on p62 and ATG7. This assay will help in elucidating the spatio-temporal dynamics and control mechanisms underlying aggregate clearance by the autophagy–lysosomal system.

---

[1] Division of Cell Biology, Department of Biology, Faculty of Science, Utrecht University, Padualaan 8, 3584 CH Utrecht, The Netherlands. [2] Department of Cell Biology, University of Groningen, University Medical Center Groningen, A. Deusinglaan 1, 9713 AV Groningen, The Netherlands. Correspondence and requests for materials should be addressed to L.C.K. (email: l.kapitein@uu.nl)

Macroautophagy (henceforth termed autophagy) is a degradation pathway that is essential for maintaining cellular homeostasis. Autophagy can either non-selectively target parts of the cytoplasm (bulk autophagy) or selectively eliminate superfluous or damaged organelles, invading pathogens, or aggregated proteins[1]. Autophagy substrates are first sequestered by a double-membrane autophagosome, which subsequently fuses with a lysosome to deliver the engulfed cargo into the hydrolytic interior of this degradative organelle. Misregulation of autophagy has been implicated in a multitude of diseases, including cancer and neurodegeneration[2,3].

Since autophagic events are rare under basal conditions, their study often requires active induction of the process. Classically, nutrient starvation or disruption of metabolic signaling by rapamycin have been used to trigger bulk autophagy. To induce selective autophagy, more recent work has attempted to trigger cargo-specific signaling pathways. For instance, recruitment of PINK1 to mitochondria triggers mitophagy to some extent[4], while overexpression of peroxisome proteins fused to ubiquitin has been used to stimulate pexophagy[5]. Other approaches have relied on damaging mitochondria using small molecules[4] or photodestruction[6,7] to induce mitophagy. In addition, xenophagy, the autophagy of intracellular pathogens, has been studied upon cell invasion by bacteria[8]. To study aggrephagy, the selective autophagy of aggregates, one could imagine introducing protein aggregates that might subsequently become cleared by autophagy. However, simply introducing aggregation-prone proteins precludes temporal control over clearance and might negatively affect cellular health and disrupt autophagic pathways[9,10]. For example, expanded polyQ proteins have been shown to interfere with polyQ-based protein–protein interactions important for autophagy regulation[9].

Recently, two inducible aggregate-forming systems have been described that rely on either the unshielding of destabilization domains[11] or the local concentration of intrinsically disordered proteins[12]. It remains unclear, however, whether these aggregates are selectively cleared through autophagy and can be used to study aggrephagy. We thus set out to develop an inducible aggregation system that allows monitoring of aggrephagy and studying the underlying principles. We previously used a chemically induced dimerization approach to create small fluorescent protein particles (particles induced by multimerization (PIMs)) to examine motor protein behavior[13]. PIMs were generated by transfection of a construct that encoded for mCherry fused to an array of FKBP12 domains (mCherry-PIM). This array comprised two repeats of FKBP, a domain that can be coupled to a FRB domain by addition of the rapamycin-analog AP21967 (referred to as rapalog1 hereafter), and four repeats of FKBP*, a variant domain that homodimerizes upon addition of the rapamycin-analog AP20187 (rapalog2 hereafter)[14]. Upon addition of rapalog2, multimerization of the FKBP* repeats concentrates the protein to form mCherry-PIM clusters, to which FRB-fused motor proteins could be recruited by addition of rapalog1. While forced motor recruitment indeed induced rapid motility of the PIMs, we noted that at longer time-scales (>30 min) PIMs would also spontaneously move and accumulate in the perinuclear space. This behavior closely resembles the reported behavior of not only protein aggregates[15], but also autophagosomes and lysosomes[16,17] and thus suggests that these clusters could be a substrate for aggrephagy.

Here we develop PIMs as a tool to study aggrephagy. Upon induction of multimerization, the clusters recruit ubiquitin, p62, and LC3 before being delivered to lysosomes. Moreover, use of a dual fluorescent tag allows for the direct observation of delivery to the lysosome. Using flow cytometry and fluorescence microscopy, we show that efficient cluster delivery to the lysosome depends on p62 and ATG7.

## Results

### PIM aggregates can be used to probe autophagic degradation.

We first ensured that the used rapamycin analogs did not impinge upon the natural target of rapamycin, mammalian target of rapamycin (mTOR) kinase, a master regulator of nutrient sensing and autophagy signaling. Indeed, treatment of cells with rapamycin, but not rapalog1 or rapalog2, strongly inhibited phosphorylation of the mTOR substrate p70S6K (Supplementary Fig. 1a). Moreover, rapalog2 did not have an effect on basal autophagy (Supplementary Fig. 1b-d). Next, we optimized the PIM construct for measuring autophagic flux by adding an enhanced green fluorescent protein (EGFP) fluorophore, resulting in a final construct comprised of four FKBP* domains for homodimerization, an EGFP and mCherry fluorophore, and two FKBP domains (mCherry-EGFP-PIM, Fig. 1a). The use of a tandem tag to monitor autophagic sequestration has been previously used to monitor the fate of p62 bodies and the maturation of autophagosomes[18,19]. As EGFP fluorescence, unlike mCherry fluorescence, is quenched under acidic conditions, the addition of EGFP enabled the PIM construct to function effectively as a pH sensor[19,20]. Therefore, if PIMs would be cleared through autophagy, their fluorescence should change from yellow (EGFP and mCherry positive) to red (mCherry positive) in the acidic lysosomal lumen (Fig. 1a). This effect is reinforced by the fact that EGFP is more sensitive to lysosomal degradation than mCherry[19,20].

Upon expression in HeLa cells, mCherry-EGFP-PIM localized diffusely in the cytosol and formed yellow clusters upon addition of rapalog2 (Fig. 1b, Supplementary Fig. 2a). Cells formed on average $77 \pm 16$ clusters (mean ± s.e.m. at 1 h after rapalog2 addition, $n = 11$ cells, Fig. 1f) with stably incorporated subunits, as shown by the limited PIM fluorescence recovery after photobleaching (Supplementary Fig. 2c). Differential detergent extraction assays revealed that these PIM clusters were soluble in 1% sodium dodecyl sulfate (SDS) but insoluble in 2% of the milder detergent Triton X-100 after induction of clustering with rapalog2 (Supplementary Fig. 2e,f). After formation, many PIM clusters accumulated in the perinuclear region (Supplementary Fig. 2b) in accordance with existing literature on aggregate behavior[15,21]. Importantly, live-cell imaging revealed that clusters started switching from yellow (EGFP and mCherry positive) to red (mCherry only) approximately 2 h after their formation (Fig. 1b, Supplementary Fig. 3a, Supplementary Movie 1). Individual clusters quickly lost 70% of EGFP fluorescence, while mCherry fluorescence remained stable (Fig. 1c, d, Supplementary Movie 2). This selective loss of EGFP fluorescence suggested a pH drop that would be consistent with transfer of PIMs into lysosomes. Indeed, red clusters colocalized with the lysosome marker LAMTOR4 (Fig. 1e). Thus, within hours upon formation, the inducible protein clusters are transferred into the lysosome, which can be followed by loss of EGFP fluorescence.

To monitor the autophagic flux in individual cells, PIM-expressing HeLa cells were imaged for 16 h after cluster formation. To quantify the progression of cluster clearance, we established an algorithm to automatically detect particles in the mCherry channel and measure the corresponding EGFP and mCherry intensities. The EGFP/mCherry ratio was normalized to the average ratio of clusters in the first frames after formation and used as a measure for lysosomal transfer, with a value <0.3 taken as evidence for entry into lysosomes. We used this algorithm to analyze the behavior of clusters in cells that showed clearance (55% of cells). After initial formation of PIM clusters, the number of yellow clusters reduced from $71 \pm 15$ to $36 \pm 7$ at 16 h after cluster induction (Fig. 1f). This decrease is partially caused by merging of smaller yellow clusters into bigger structures as we initially also observed a drop in the total number of clusters

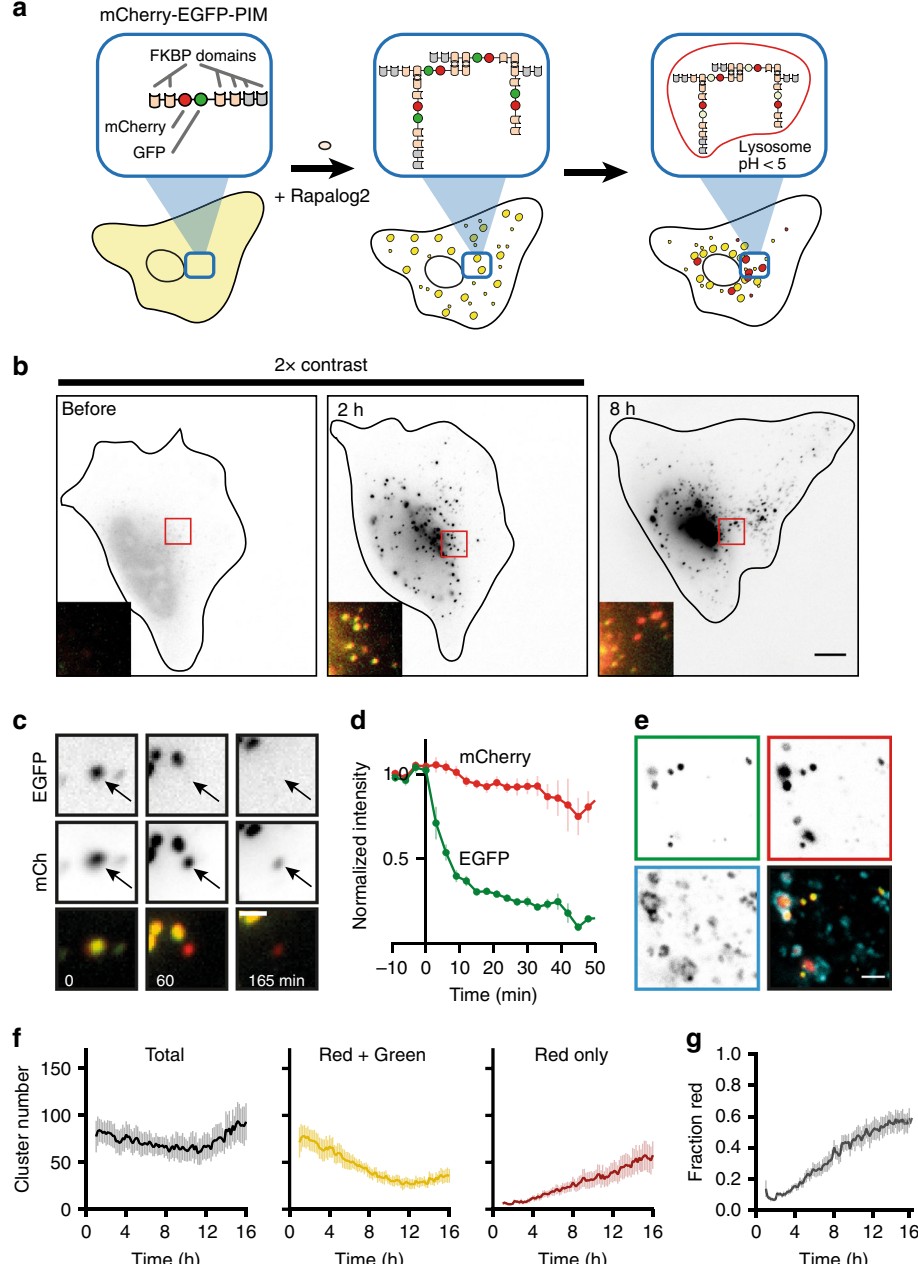

**Fig. 1** Cluster formation and degradation in cells. **a** Assay: rapalog2-induced homodimerization induces red/green PIMs (particles induced by multimerization) that become red only upon entry into the lysosome. **b** HeLa cell expressing PIM construct showing cluster formation and degradation. Inverted contrast gray scale images show mCherry channel while inserts show merged image. The image contrast in the left and middle panel are 2× that of the right panel. Merged images of the full cell can be found in Figure S2. **c** Time-lapse images of an individual cluster. Arrow tracks cluster showing color conversion. **d** Normalized mCherry (red) and EGFP (green) fluorescent intensity of individual clusters versus time (mean ± s.e.m. $n = 9$ clusters). $t = 0$ marks entry into lysosome. **e** Immunofluorescence images of clusters in HeLa cells 24 h after cluster formation. Panels show endogenous staining of LAMTOR4 (cyan panel), clusters in mCherry (red panel) and EGFP channel (green panel). **f** Quantification of 16 h live-cell imaging of individual cells. Average number of total cluster (black), red and green clusters (yellow) and red-only clusters (red) per cell (mean ± s.e.m. $n = 11$ cells from 3 independent experiments). The x axis indicates time after rapalog2 addition. **g** Average fraction of red clusters of total clusters. Data are mean ± s.e.m. $n = 11$ cells from 3 independent experiments. Scale bars, 10 µm (**b**) and 2 µm (**c**, **e**)

detected (Fig. 1f). Nevertheless, the number of red-only clusters strongly increased from $6 \pm 1$ to $58 \pm 14$ at 16 h (Fig. 1f), indicating an average autophagic flux of 3.4 clearance events per hour per cell. The fraction of red-only clusters was $40 \pm 5\%$ at 8 h and $59 \pm 6\%$ after 16 h (mean ± s.e.m., $n = 11$ cells, Fig. 1g). We also quantified the EGFP/mCherry ratio of the integrated intensity of all detected clusters in a cell over time, which revealed that, after 16 h, the intensity ratio had dropped from 1 to 0.5

(Supplementary Fig. 3c). The exact behavior of PIM clusters varied between cells. In most cells (70%), direct clearance of smaller clusters was observed (Supplementary Fig. 3d). In a few cells (3%), PIM clusters first merged to larger perinuclear clusters, followed by degradation of parts of these clusters at later stages (Supplementary Fig. 3e). Finally, 27% of cells showed a combination of these types of behavior with initial clearance of smaller clusters followed by merging and subsequent degradation

of larger structures. Despite these differences, our assay can successfully be used to monitor autophagic flux and observe the dynamics of the autophagy process.

**PIM clusters are recognized by selective autophagy markers**. Lysosomal delivery takes place via several pathways[22]. In selective autophagy, targets are recognized by autophagy receptors, such as p62 and/or NBR1, that initiate membrane recruitment through interaction with LC3 and other autophagy-related (ATG) proteins[18,23–25]. Recognition of ubiquitinated aggregates is typically mediated by p62, but ubiquitin- and p62-independent aggrephagy mechanisms have also been described[26]. To test whether PIM clusters were cleared by selective autophagy, the colocalization of PIMs with marker proteins of different steps in the pathway was examined. In the absence of rapalog2, we did not find a significant effect of PIM expression on the localization of the markers that we examined (Supplementary Fig. 4). However, upon PIM cluster formation, ubiquitin, p62, and NBR1 adopted a more distinct punctate pattern that colocalized with the PIM clusters (Fig. 2a–c). Yellow clusters colocalized with these markers more frequently than red clusters, consistent with degradation of ubiquitin, p62, and NBR1 in the lysosome. To test whether the PIM protein was directly ubiquitinated, we expressed the construct in HEK cells and performed His-ubiquitin pulldowns. Indeed, we observed direct ubiquitination of the PIM protein, although not in a rapalog2-dependent manner (Supplementary Fig. 2d). The rapalog2 independence could be caused by the high expression of the PIM construct in HEK cells, which resulted in aggregation without the addition of rapalog2. The autophagosome marker LC3 decorated only a small subset of clusters (Fig. 2d) suggesting rapid fusion between autophagosomes and lysosomes. Indeed, when fusion was blocked by treatment with the specific vacuolar-type $H^+$-ATPase inhibitor Bafilomycin A1[27], increased colocalization with LC3 was observed (Fig. 2e). Finally, LAMTOR4 colocalized with most red clusters (Fig. 2f). Together, these results show that PIM clusters are recognized by markers of the aggrephagy pathway and delivered to the lysosome.

**PIM cluster degradation (partially) depends on p62 and ATG7**. To examine larger numbers of cells in different conditions without the need for live-cell imaging, we fixed cells after 1 or 8 h following the addition of rapalog2. We analyzed the EGFP/mCherry ratio of all clusters in the cells imaged using the algorithm described above and plotted these in a histogram. Clusters were called red only when the normalized ratio was <0.3. In wild-type (WT) HeLa cells, the percentage of red-only particles was 4% and 21% after 1 and 8 h, respectively (Fig. 3a). This value is lower than the one obtained from live-cell analysis, because in the latter we excluded cells without any clearance and low-expressing cells that only form clusters several hours after induction (Supplementary Fig. 3b). Importantly, treatment with Bafilomycin A1 resulted in a complete loss of red-only clusters at all time points, confirming that PIM conversion depends on active lysosomes (Fig. 3b). These results demonstrate that our method of aggregate induction, in combination with automated image analysis, can be used to rapidly monitor the flux of aggregate turnover by autophagy.

To investigate the importance of p62 in the clearance of induced clusters, this autophagy receptor was deleted in HeLa cells (p62 knockout (KO)) by CRISPR-Cas9-mediated genome editing. Complete knockout of p62 was validated by loss of p62 staining on western blot (WB) and in immunofluorescence (Supplementary Fig. 5b,c). Importantly, starvation-induced bulk autophagy was not affected in these cell lines as shown by

lipidation of LC3 (Supplementary Fig. 5d,e). In p62KO cells, the population-wide degradation of PIM clusters was impaired, demonstrating that efficient PIM clearance is p62 dependent (Fig. 3c, Supplementary Fig. 5a). Furthermore, to examine whether PIM clusters undergo canonical autophagy, we tested clearance in ATG7KO U2OS cells. Knockout of ATG7 was confirmed by WB (Supplementary Fig. 6c). Consistent with previous reports[28,29], knockout of ATG7 resulted in a defect in p62 degradation by starvation-induced autophagy as well as impairment in LC3 lipidation (Supplementary Fig. 6d,e). Although PIM clearance was generally slower in U2OS than in HeLa cells, we observed a reduction in PIM clearance at 16 h after aggregate induction from 18% in U2OS WT to 8% in ATG7KO (Supplementary Fig. 6a,b). Thus, while PIM clusters are largely cleared through the canonical ATG7-dependent pathway, they can, to a lesser extent, be cleared in the absence of ATG7.

To examine the relationship between aggregate size and clearance, we analyzed the size distribution of the clusters in fixed HeLa cells. In WT cells, the size distribution of yellow and red clusters was similar, indicating no preference for the clearance of aggregates of specific sizes (Fig. 3d, e). In p62KO cells, however, we observed a clear difference in the size distribution of yellow and red clusters (Fig. 3d). Here the red clusters were typically smaller, which suggests that p62KO cells are mainly impaired in the clearance of bigger clusters. Together, these data demonstrate that most PIM clusters undergo the complete process of p62-dependent autophagy.

**Fluorescence-activated cell sorting (FACS) analysis shows robust PIM clearance**. To determine whether our assay is compatible with high-throughput approaches, we analyzed mCherry-EGFP-PIM-expressing cells using FACS. As expected, based on the drop in EGFP/mCherry ratio of the integrated intensity of all detected clusters in the live-cell analysis (Supplementary Fig. 3c), we indeed observed a clear shift in EGFP/mCherry ratio over time in a large fraction of cells (Fig. 4a, d). Histograms of the EGFP/mCherry ratios for different time points show the emergence of a second peak at lower ratios (Fig. 4a), which was largely absent upon treatment of cells with Bafilomycin A1 or in p62KO cells (Fig. 4b, c, e, Supplementary Fig. 7). These results demonstrate that our assay facilitates robust detection of aggrephagy using FACS, which enables high-throughput screening approaches.

## Discussion

By using inducible protein clustering, we have established a tool to induce and monitor the turnover of aggregates by autophagy in living cells. Upon formation, clusters are ubiquitinated and recruit p62, NBR1, and LC3 before final degradation in lysosomes. Efficient clearance depends on p62 and ATG7, suggesting that our assay probes a selective type of autophagy, i.e., aggrephagy. p62 appears more important for the clearance of larger aggregates, which could suggest that multiple types of autophagy are involved. These findings are consistent with previous studies which suggest that clearance of smaller aggregates might be dependent on basal autophagy, while larger aggregates need induction of autophagy for their turnover[30]. The clearance observed in the absence of ATG7 could be mediated by alternative macroautophagy[31]. Alternatively, canonical autophagy in an ATG-conjugation-independent mechanism has also been described[32], although it has remained unclear whether such ATG-independent autophagosome-like structures can mature into autolysosomes. Therefore, it would be interesting to explore the mechanisms involved in the residual clearance of aggregates in ATG7KO cells.

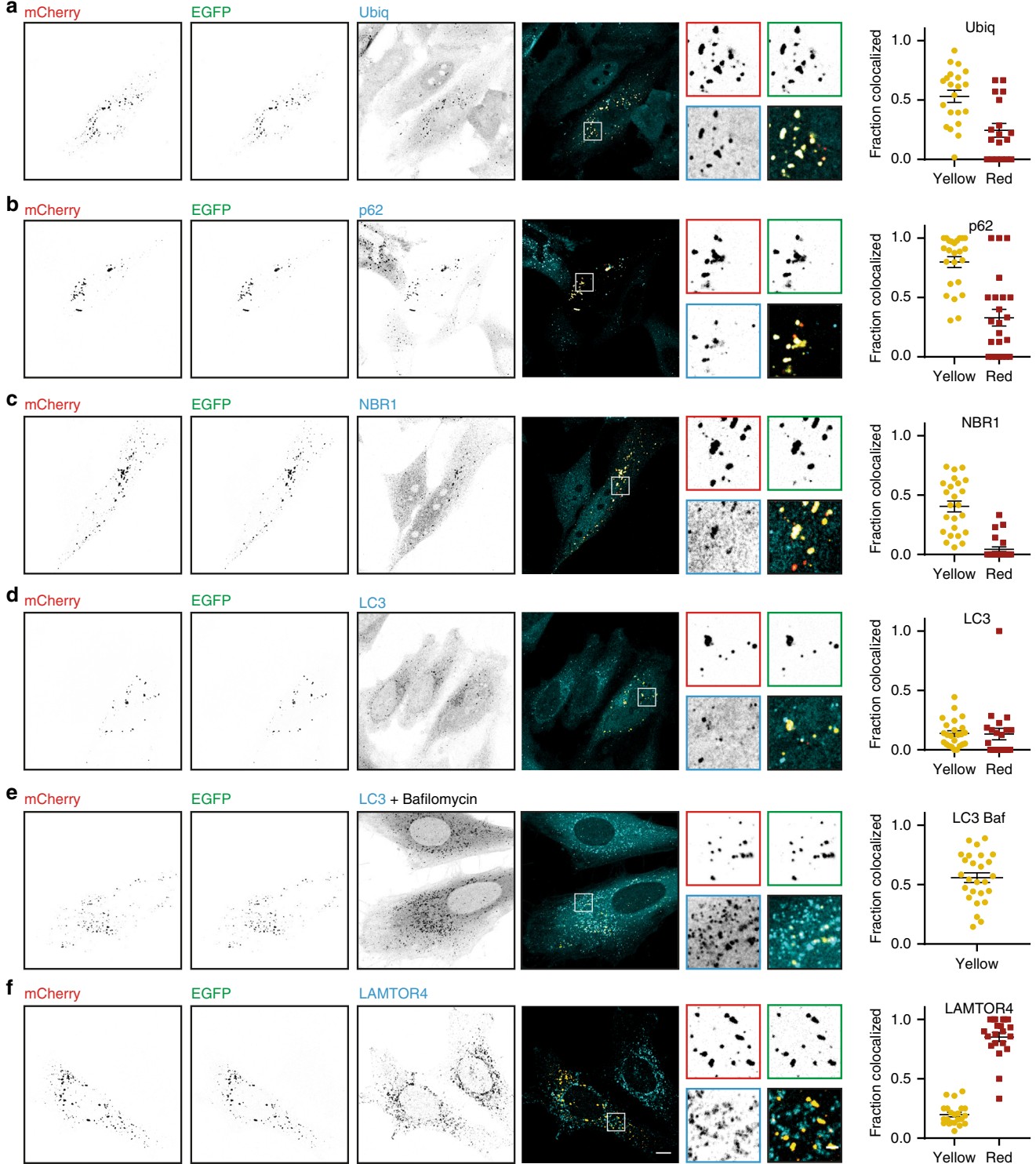

**Fig. 2** Colocalization of PIM clusters with the autophagic machinery. Immunofluorescence images of representative HeLa cells 8 h after cluster induction by rapalog2 addition. mCherry (red), EGFP (green), and endogenous staining (cyan) of ubiquitin (**a**), p62 (**b**), NBR1 (**c**), LC3 (**d**, **e**), and LAMTOR4 (**f**) shown in inverted contrast. For **e**, HeLa cells were treated with 200 nM Bafilomcyin A1 for 8 h. Scale bars, 10 µm. Zooms show individual clusters, scale bar 2 µm. Plots show the fraction of yellow (yellow circles) and red (red squares) clusters colocalized with the marker per cell. Mean ± s.e.m. from 2 independent experiments with $n = 21$, 24, 24, 24, 26, and 22 cells for **a**–**f**, respectively. For images of PIM-expressing cells without rapalog2 treatment, see Supplementary Figure 4

Furthermore, we observed various types of behavior in different cells. Typically, cells with low PIM–protein expression form small clusters that are rapidly cleared. Higher expressing cells initially show some clearance of smaller aggregates, but also form bigger clusters by merging multiple PIMs. These bigger clusters resemble aggresomes, which have been suggested to form as protective mechanism when the clearance burden is too high[15]. Future work should be aimed at identifying the factors involved

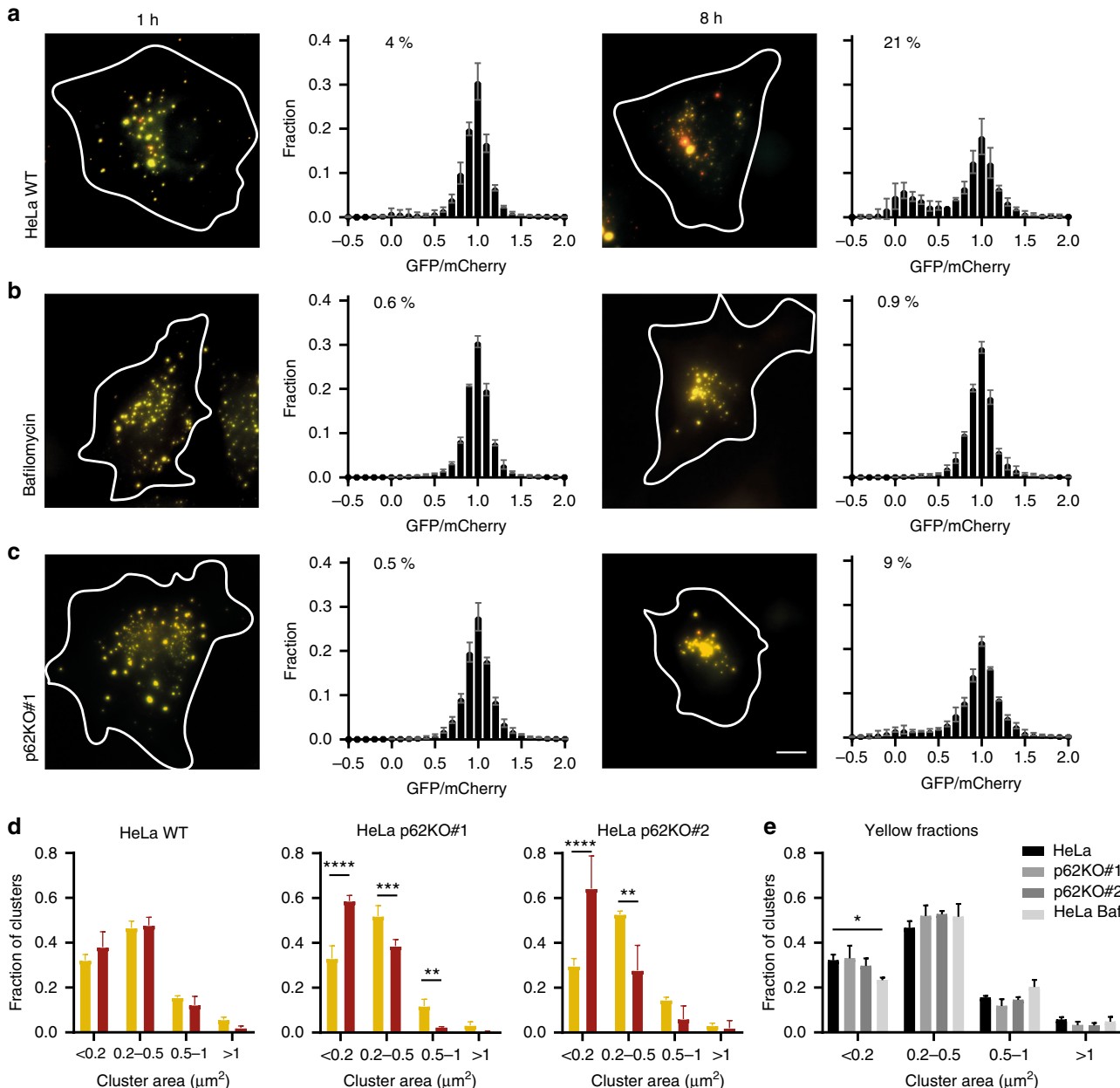

**Fig. 3** Cluster degradation at different time points. Distribution of normalized EGFP/mCherry ratios of clusters at different time points after cluster formation in **a** HeLa WT cells treated with 0.2% DMSO, **b** HeLa WT cells treated with 200 nM Bafilomycin A1, and **c** HeLa p62KO#1 cells. Each histogram represents >3000 clusters. Mean ± s.e.m from 3 independent experiments with 40–60 cells per condition. Percentage of red clusters is indicated as the average fraction of clusters with an EGFP/mCherry ratio <0.3 of 3 independent experiments. **d** Size distribution of clusters analyzed in **a–c** and Fig. S5. Indicated are the fraction of EGFP- and mCherry-positive clusters (yellow, ratio EGFP/mCherry >0.3) and mCherry-positive clusters (red, EGFP/mCherry <0.3) that belong to different size categories. Data are expressed as mean ± s.d. ($n = 3$). **$P \leq 0.01$, ***$P \leq 0.001$ ****$P \leq 0.0001$ (one-way ANOVA with Sidak's post-hoc test). **e** Size distribution of yellow clusters analyzed in **a–c** and Fig S5a. Data are expressed as mean ± s.d. ($n = 3$). *$P \leq 0.05$ (one-way ANOVA with Dunnett's post-hoc test). Scale bar 10 μm

in the different types of clearance and at exploring how cells sense their aggregate burden and determine their clearance strategy. In addition, the efficient clearance of these artificial aggregates raises intriguing questions about their recognition. It has previously been suggested that the dynamic properties of an aggregate's surface, rather than the nature of the aggregate as such, are what determines its fate[30,33]. Our system could aid in dissecting the cues that mediate effective recognition and targeting for destruction. Our assay will also be highly relevant to further explore the dynamics and spatiotemporal regulation of selective autophagy. For example, the timing of different steps can be

monitored in relation to the positioning and the degradation status of the clusters. Furthermore, co-expression of pathological aggregates could reveal which steps in the process are affected in different proteinopathies. Finally, as FACS analysis showed robust clearance in our system, it is possible to use our PIM assay as a screening approach for potential modulators of aggrephagy.

## Methods
**Construct**. The PIM construct used in this study was cloned into the mammalian expression vector pβactin. The construct consists of four FKBP homodimerization domains with sequence variation, mCherry, EGFP, and two FKBP

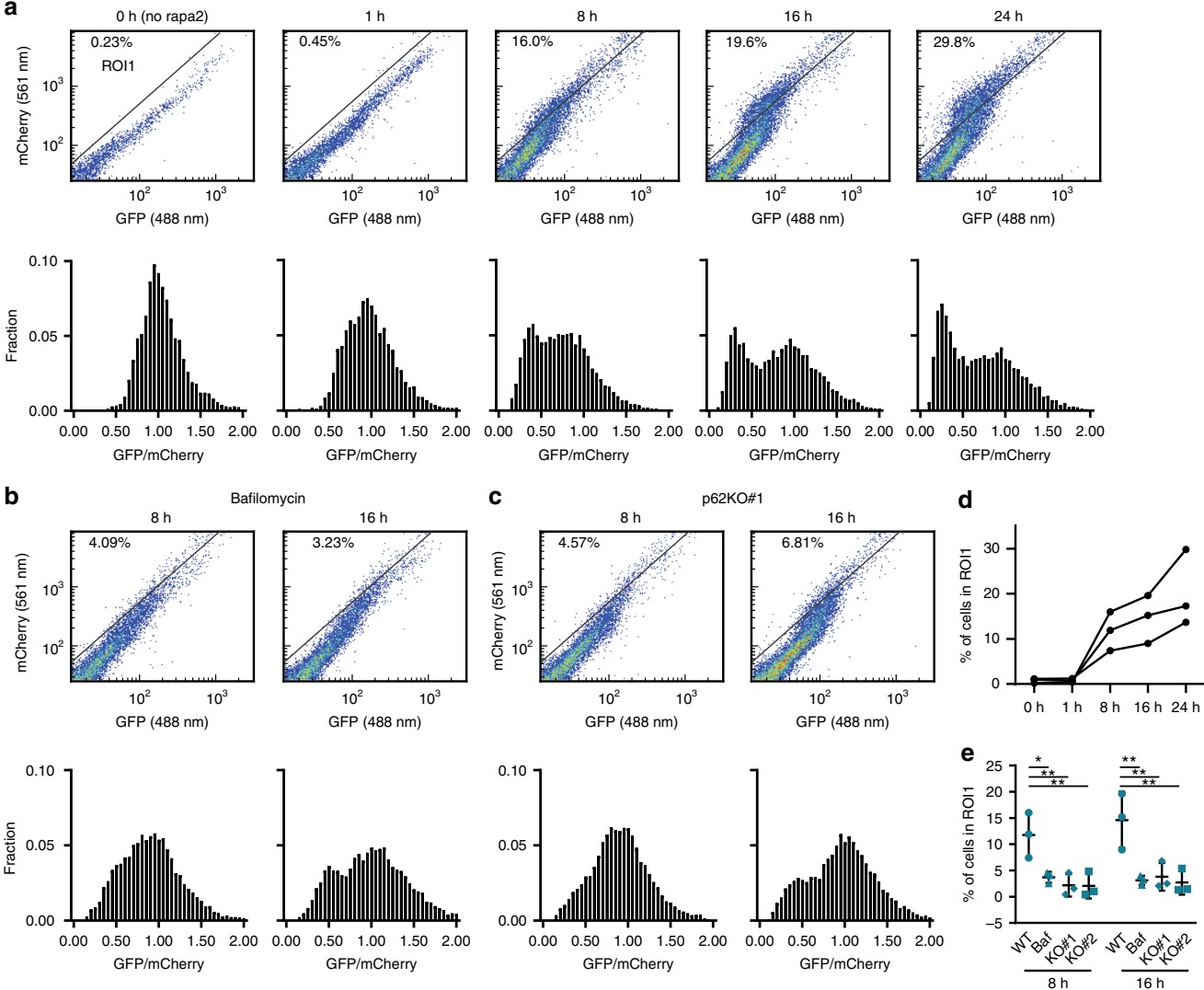

**Fig. 4** Population-wide PIM degradation using flow cytometry. Cells were treated with rapalog2 and analyzed by FACS for a shift in GFP/mCherry ratio. Scatter plots and matching histograms of GFP/mCherry ratio are shown. **a** Representative data for HeLa WT cells at different time points after rapalog2 addition. **b** Representative data for HeLa WT cells at 8 and 16 h after rapalog2 addition treated with Bafilomycin A1. **c** Representative data for p62KO#1 at 8 and 16 h after rapalog2 addition. **d** Percentage of HeLa WT cells in ROI1 at different time points after rapalog2 addition. Data from three independent experiments. **e** Percentage of cells in ROI1 for HeLa WT, Bafilomycin A1-treated cells, and p62KO#1 and p62KO#2 at 8 and 16 h after rapalog2 addition. Mean ± s.d. from three independent experiments. One-way ANOVA reveals $F = 8.516$, $P = 0.0072$ for 8 h and $F = 9.422$, $P = 0.0053$ for 16 h. Dunnett's post-hoc test: $^*P \leq 0.05$, $^{**}P \leq 0.01$

heterodimerization domains. The first two FKBP homodimerization domains contain three mutations (V24E, Y80C, and A94T) that were found to aid multimerization. The construct and full sequence can be found on Addgene (#111758).

**Cell culture and transfection.** HeLa, U2OS, and HEK293T cells were cultured in Dulbecco's modified Eagle's medium containing 10% fetal calf serum and Penicillin/Streptomycin. Cells were maintained at 37 °C and 5% $CO_2$. Cells were regularly tested for mycoplasma contamination using MycoAlert Mycoplasma Detection Kit (Lonza). Cells were plated on 18-mm diameter coverslips 1–3 days before transfection. Cells were transfected using Fugene6 transfection reagent (Roche) according to the manufacturer's protocol. Experiments were started 1 day after transfection. HeLa and HEK293T cells were purchased from ATCC, and U2OS cells were a gift from G. Strous, University Medical Center Utrecht, the Netherlands.

**Generation of KO cell line using CRISPR/Cas9 gene editing.** To generate the p62 HeLa knockout cell line, CRISPR guide RNAs (gRNAs) were chosen that target exon 3, which is the first exon present in all isoforms of the protein. A previously described CRISPR/Cas9 system was used to generate p62KO line[34,35].Briefly;

oligonucleotides (IDT DNA) containing sgRNAs were cloned into PX459 v2 vector (Addgene #62988). HeLa cells were transfected using Fugene6 and put on puromycin (1 μg/ml) selection the next day. After 2–3 days, cells were taken off selection and allowed to expand. Different sgRNAs were designed and the efficiency of knockout was assessed on the polyclonal population by immunoblotting. Polyclonal lines that showed most efficient reduction in protein level were plated in 96-well plates. Single clones were allowed to expand into 12-well plates before screening for knockout lines by immunoblotting and immunofluorescence. Mouse anti-p62 (Abnova, #H00008878-M01, 1/2000) was used as primary antibody.

To generate ATG7KO U2OS cells using the CRISPR/Cas9 system, guides targeting exon 1 of *ATG7* were designed using optimized CRISPR design (http://crispr.mit.edu/). Guides were cloned into pX458 plasmid (Addgene #48138) allowing expression of Cas9 along with GFP. U2OS were transfected and 48 h later clonally sorted based on GFP expression. Clones were then sequenced and protein expression was assessed by WB. The following primary antibodies were used: rabbit anti-ATG7 (Cell Signaling Technology, #2631S, 1/1000) and mouse anti-actin (Merck, #MAB1501, 1/5000).

**Immunoblotting of mTOR kinase activity and basal autophagy.** For immunoblotting, Hela cells were treated with 500 nM rapamycin, 500 nM rapalog1, and

500 nM rapalog2. After 4 h, cells were washed 2× in ice-cold phosphate-buffered saline (PBS), lysed in Laemmli buffer, and processed for SDS-polyacrylamide gel electrophoresis (PAGE). Proteins were transferred to nitrocellulose membranes for immunoblotting. Blots were blocked in 5% milk PBST (0.1% Tween20 in PBS) and incubated overnight at 4 °C with primary antibody and for 1 h at room temperature with IRDye-conjugated secondary antibodies (LICOR). Signal was visualized and scanned on an Odyssey imaging system. The following primary antibodies were used: rabbit anti-phospho-S6K(Thr389) (CST, #9205, 1/5000), rabbit anti-S6K (CST, #2708, 1/5000), mouse anti-alpha-tubulin (Sigma, #T5168, 1/20000).

For determination of the effect of rapalog2 on basal autophagy, HeLa cells were treated with 500 nM rapalog2 for 4 h and/or 200 nM Bafilomycin A1. Cells were lysed and processed for WB. Primary antibodies used were mouse anti-p62 (Abcam, #ab56416, 1/2000), mouse anti-ubiquitin (Enzo, #BML-PW8810, 1/2000), and rabbit anti-GAPDH (Sigma, #G9545, 1/5000). All uncropped WBs are available in Supplementary Figure 8.

**LC3 immunoblotting**. U2OS WT, ATG7KO, HeLa WT, and HeLa p62KO were seeded into a 12-well plate for 24 h. Cells were then washed twice with PBS and treated for 2 h with EBSS (ThermoFischer, #24010043) and/or 200 nM Bafilomycin A1. For LC3 immunoblotting with rapalog2 treatment, HeLa cells were treated for 4 h with 500 nM rapalog2. Finally, cells were lysed and processed for WB.

Primary antibodies used were: rabbit anti-LC3 (Novus Biologicals, #NB600-1384, 1/1000), mouse anti-p62 (Abcam, #ab56416, 1/1000), mouse anti-actin (Merck, #MAB1501, 1/5000), and mouse anti-tubulin alpha (Sigma, #T5168, 1/20,000). Goat anti-mouse and goat anti-rabbit secondary antibodies conjugated to Alexa Fluor 680/800 were used for visualization and purchased from ThermoFischer Scientific.

**Fluorescence microscopy**. Live-cell imaging was performed on a Nikon Eclipse TE2000E. An incubation chamber (Tokai Hit; INUG2-ZILCS0H2) was used that was mounted on a motorized stage (Prior). Coverslips were mounted in metal imaging rings immersed in medium. During imaging, cells were maintained at 37 °C and 5% CO$_2$. Cells were imaged every 3 or 10 min for ~8 and 16 h, respectively, using a ×40 oil immersion objective (Plan Fluor, NA 1.3, Nikon) and a Coolsnap HQ2 CCD camera (Photometrics). A mercury lamp (Osram) and filter wheel containing ET-GFP (49002, chroma) and ET-mCherry (49008, chroma) emission filters were used. To start cluster formation, 500 nM Rapalog2 (Clontech, AP20187) was added. Rapalog2 was washed out before the start of image acquisition in all experiments that were used for quantifications. To perform 16 h imaging, rapalog2 was added to cells for 45 min after which coverslips were mounted in medium without Phenol red and rapalog2 in an imaging ring with coverslip on top to prevent medium evaporation. Imaging was started 1 h after rapalog2 addition.

For fixed cell analysis of PIM cluster degradation, HeLa WT or p62KO cells were transfected 24 h before starting the experiment. One hour before the start of aggregation, 0.2% dimethyl sulfoxide (DMSO) or 200 nM Bafilomycin A1 was added to WT cells. Rapalog2 500 nM (AP20187, Clontech) was added to induce aggregate formation, and after 1 h, medium was replaced by fresh medium with DMSO/Bafilomycin for WT cells. Aggregation was started at different time points (8 and 1 h before fixation) so that all cells could be fixed simultaneously. Cells were fixed using 4% paraformaldehyde (PFA) and were mounted using Prolong Diamond. Images were taken on the same Nikon Eclipse TE2000E set-up using a ×60 oil immersion objective (Plan APO, NA 1.4, Nikon). Ten different positions were picked by searching in the red channel for cells that showed an average levels of aggregates. A relative z-stack was taken from −1.5 to 1.5 μm with 0.5 μm steps. Imaging settings used were consistent throughout experiments. These images were also used for quantification of PIM localization.

**Image processing and analysis**. For analysis of EGFP loss in individual clusters, a region of interest (ROI) was placed around the cluster in every frame and fluorescent intensity was measured in both channels. Background intensity was subtracted and the EGFP and mCherry intensities were normalized to the average intensity of the first five frames. Subsequently, the EGFP/mCherry ratio was calculated. The first frame at which the ratio drops <0.9 was put at $t = 3$ min (first frame) so that start of degradation was synchronized. Frames at which the tracked cluster was out of focus or when other clusters were too close were not included in the analysis and were given no value.

Live-cell cluster clearance over 16 h was analyzed by first placing an ROI around the whole cell. ROIs per cell were adapted as cells migrated throughout the imaging session. Only cells that showed at least some yellow to red conversion and could be followed for at least 16 h were included. Cells that showed late aggregation (i.e., after several hours), died, or divided during the imaging session or that at some point lost focus were discarded. In addition, cells in which all PIMs converged to one single cluster were not included. Cluster detection, colocalization, and quantification was performed using ComDet v.0.3.7 plugin for ImageJ (https://github.com/ekatrukha/ComDet). In short, at first particles were detected in mCherry channel and for each detected spot a bounding rectangular area around it was recorded. The integrated intensity of a spot was calculated as a sum of pixels inside the rectangle corrected for the background value, estimated as an average of

pixels comprising the rectangle's perimeter. The same rectangles were used to quantify intensity in the GFP channel. ComDet v0.3.7 Plugin was used to detect particles >4 pixels in the mCherry channel using a signal-to-noise ratio (SNR) of 20. Data were transferred to Microsoft Excel and values were normalized per cell by dividing ratios by average ratio of clusters in the first 5–10 frames with a value >0. Clusters with a ratio <0.3 were regarded as red clusters. All other clusters were categorized as red and green positive (yellow). The fraction of red clusters was calculated by dividing the number of red particles by the total number of particles at that specific time point. Averages were indicated per cell. Graphs were plotted using the GraphPad PRISM7 software.

For analysis of fixed samples, cells that were not entirely in field of view were discarded from analysis. Also cells with high aggregate levels resulting in no visible individual aggregates were discarded. An average z-projection was made from the stack using ImageJ (NIH). An ROI was placed around the cell and ComDet v0.3.7 Plugin was used to detect particles >4 pixels in the mCherry channel using an SNR of 5. Data were transferred to Microsoft Excel. Values were normalized by first fitting with a single or double Gaussian distribution using the GraphPad PRISM7 software. The mean of the peak with the highest ratio, which represents EGFP- and mCherry-positive clusters, was used for normalization. Frequency distribution graphs were plotted using the GraphPad PRISM7 software. Final graphs represent data from 40 to 60 cells from three independent experiments. The percentage of red clusters is calculated as average percentage of the three independent experiments of clusters with an EGFP/mCherry ratio <0.3.

**Immunofluorescence cell staining, imaging, and antibodies**. Clusters were formed in HeLa cells (1 day after transfection) by addition of 500 nM rapalog2. Rapalog2 was washed out after 1 h and cells were fixed 7 h later. Cells were fixed at room temperature for 10 min with 4% PFA. Cells were washed in PBS, permeabilized using 0.2% Trition-X100, and blocked using 3% bovine serum albumin (BSA) in PBS. Cells were incubated overnight at 4 °C in 3% BSA PBS containing primary antibody. Next day, cells were washed using PBS and incubated for 1 h at room temperature with secondary antibody in 3% BSA PBS. Cells were washed in PBS and mounted using Prolong Diamond (Thermo Fischer). Confocal images were taken on Leica TCS SP8 STED 3× microscope using HC PL AP ×100/1.4 oil STED WHITE objective. Analysis was performed using the ImageJ software.

The following primary antibodies were used: rabbit anti-LC3 (MBL, #PM036, 1/200), mouse anti-p62 (Abnova, #H00008878-M01, 1/500), rabbit anti-NBR1 (Novus, #NBP1-71703, 1/500), rabbit anti-LAMTOR4 (CST, #12284S, 1/500), and mouse anti-ubiquitin (Enzo, #BML-PW8810, 1/500).

**Fluorescence recovery after photobleaching (FRAP)**. For FRAP analysis of aggregate stability, clusters were imaged using a ×100 objective (Apo TIRF, 1.49 NA, Nikon) on a Coolsnap Myo camera (Photometrics). A FRAP scanning head was used (FRAP L5 D-CURIE, Curie Institute) to bleach clusters using a 568-nm laser. Clusters were imaged for 180 frames every 5 s to monitor recovery. Intensity values were determined using ImageJ and normalized with pre-bleaching intensities.

**Protein extraction**. One day after transfection, cells were treated with rapalog2 for 1 h. After 4 h, cells were washed once in cold PBS and lysed in 200 μl 1% Triton X-100 in PBS or 1% SDS in PBS containing 1% complete protease inhibitor cocktail (Roche Applied Science). Cell lysates were scraped and sonicated briefly. Protein levels were determined using the BCA Protein Assay Kit (Thermo) and the input was equalized. For SDS-based fractionation, cell lysates were centrifuged at $20,000 \times g$ for 30 min at 4 °C. The supernatant was collected (soluble fraction) and pellets were washed 1× using PBS. Samples were prepared for WB analysis. For TX-100-based fractionation, samples were first centrifuged at $380 \times g$ to remove larger cell debris. The supernatant was collected and Triton concentration was increased to 2% after which the samples were fractionated as described for the SDS-based samples. Immunoblotting was performed using rabbit anti-GFP (Abcam, ab290, 1/5000) and mouse anti-tubulin alpha (Sigma, T5168, 1/20,000).

**His-ubiquitin pulldown**. HEK293T cells were transfected with the mCherry-EGFP-PIM construct and pMT107 (His-Ubi) using Fugene6 according to the manufacturer's protocol. PIM construct in GW1 vector with CMV promoter was used in these experiments for higher expression levels (Addgene #111759). One day after transfection, rapalog2 was added for 1 h. Four hour after aggregate formation, pulldown protocol was performed. Briefly, cells were washed in cold PBS after which cells were gently scraped in cold PBS with 10 mM NEM. Cells were pelleted by centrifugation and resuspended in pH 8.0 buffer (6 M guanidine HCl, 0.1 M NaH$_2$PO$_4$-Na$_2$HPO$_4$, 10 mM Tris-HCl pH 8.0, 25 mM DTT). After short sonication, input samples were collected after which cell lysates were incubated with cOmplete His-Tag purification Resin (Roche) in pH 8.0 buffer with 5 mM Imidazol at 4 °C overnight. The next day, resin was washed with pH 8.0 buffer with 0.05% Tween-20 and buffer pH 6.3 (8 M Urea, 0.1 M NaH$_2$PO$_4$-Na$_2$HPO$_4$, 10 mM Tris-HCl pH 6.3, 0.1% Tween-20, and 25 mM DTT). After the last wash, resin was pelleted and resuspended in sample buffer and boiled. Samples were loaded onto SDS-PAGE gels. Immunoblotting was performed using rabbit anti-RFP (Rockland,

#600-401-379, 1/2000) and mouse anti-ubiquitin (Enzo, #BML-PW8810, 1/2000) antibodies using standard immunoblotting protocol.

**Fluorescence-activated cell sorting**. FACS analysis was performed on a BD-Influx cell sorter. Measurements were made using a 488 (EGFP) and 561 (mCherry) nm laser with 520/35 nm and 610/20 nm emission filters, respectively. For each sample, 20,000–50,000 events were collected and subsequently gated for singlets and EGFP- and mCherry-positive cells. Data were analyzed using FlowJo v10 and plotted using MATLAB and GraphPad.

**Statistics**. Statistical analysis was done using GraphPad Prism v7. For analysis of variance, post testing was performed as indicated to correct for multiple comparisons. Error bars shown in the figures are standard deviation (s.d.) or standard error (s.e.m.), as stated. Sample size was not predetermined and experiments were not randomized. A normal distribution was assumed. Tests were two-tailed.

**Code availability**. ComDet v.0.3.7 plugin for ImageJ is available online (https://github.com/ekatrukha/ComDet).

### Data availability
The PIM construct has been deposited in Addgene. All relevant data are available from the authors.

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

### Acknowledgements
We are grateful to Fried Zwartkruis for reagents (pS6K and S6K antibodies), to Ger Arkestijn for help with FACS analysis, and to Wilco Nijenhuis for comments on the manuscripts. We would like to thank Harrie Kampinga and Maria Waarde-Verhagen for useful suggestions regarding aggregate characterization. This research was funded by the Netherlands Organization for Scientific Research (NWO) through an ALW-VIDI grant to L.C.K. (ALW-VIDI 864.12.008). F.R. is supported by SNF Sinergia (CRSII3_154421), Marie Skłodowska-Curie Cofund (713660), Marie Skłodowska-Curie ITN (765912), and ZonMW VICI (016.130.606) grants. L.C.K. and F.R. are also supported by a ZonMW TOP grant (91217002).

### Author contributions
A.F.J.J. and L.C.K. designed research. A.F.J.J. created reagents, performed experiments, and analyzed data. E.A.K. wrote the software for automatic particle detection and quantification. W.S. created p62 knockout cells. P.V. created and validated U2OS ATG7 knockout cells. F.R. provided advice and guidance on autophagy and contributed reagents. A.F.J.J. and L.C.K wrote the manuscript with input from the other authors. L.C. K. supervised the study.

### Additional information

**Competing interests:** The authors declare no competing interests.

