## [Peer Review File · Nature Communications]

Reviewers' comments:

Reviewer #1 (Remarks to the Author):

Janssen and colleagues describe in their manuscript a new method for probing the turnover of inducible cytosolic aggregates by selective autophagy. They exploit the chemically-induced multimerization of mCherry containing 4 FKBP domains, which forms intracellular aggregates (PIMs, particles induced by multimerization) upon treatment with the rapalog AP20187 (rapalog2), in follow their turnover. In order to follow autophagic sequestration and lysosomal degradation they modify this construct with an additional GFP-tag to monitor quenching of GFP-fluorescence in autolysosomes/lysosomes.

Upon induction with rapalog2 they observe formation of yellow PIMs, which accumulate in the perinuclear region, and importantly after long term observation show a shift to the red color, indicating quenching of the GFP fluorophore. The dynamics of cluster formation and possible sequestration by the autophagic machinery was followed both by live cell imaging and on fixed cells. In addition, the authors perform quantification of live cell imaging data of clusters and follow their fate over a 16-hour period, showing that a certain percentage of these clusters are undergoing a shift from yellow to red fluorescence, indicating accumulation in an acidic environment. To further address the question whether some of the cluster actually are delivered to the autophagic pathway the authors perform colocalization experiment against autophagic adaptor proteins such as p62 and nbr1, as well as ubiquitin, a well established marker for autophagic targeting. They also label against the most widely used autophagic marker protein LC3 and the lysosomal marker Lamtor4. They conclude from these experiments that a certain proportion of the PIMs are actually transferred to lysosomal degradation by selective autophagy. Using a CRISPR-Cas-9 mediated knock-out of p62 they further show that this happens in a p62-dependent manner.

On a general basis I would like to comment that the manuscript could be of interest for the autophagic community, however that would depend on a more thorough investigation of the turnover these inducible cluster. There are several keys issues which would need to be addressed by the authors for a clear demonstration of autophagic turnover.

There are some shortcomings on statistical significance of some experiments as noted below. With respect to reproducibility of the experiments I think it would be challenging for other labs to perform similar experiments, as many of the measurements are taken on a selection of induced clusters, however, the nature of this study somehow makes it mandatory to use certain selection criteria, especially when doing live cell imaging. So in my view this biased approach is necessary in this context.

A greater impact on a broader audience of these data in the present form is in my opinion not given.

Major points:

1. The only clear evidence for delivery to the autophagic pathway is the actual colocalization with autophagic adaptor proteins (nbr1 and p62) and the rather poor localization data with LC3 and Lamtor4. The authors state actually that "only some" red clusters colocalize with both LC3 and Lamtor4, which is surprising with regard to their interpretation of autophagic turnover. These data lack also quantification, which should be straight forward to perform in these experiments, and would make a much stronger argument. On a technical note I was surprised to read that the fixation protocol for the IF images was only a 10 min PFA fixation, followed by overnight incubation with antibody in rather harsh conditions (Tween and BSA), which from my experience might be the reason for unsatisfactory labeling results.
2. The authors should perform experiments with atg7 and atg5 siRNA-knock down to characterize the involvement of the core-autophagy machinery. Off course that would not exclude the

possibility for an atg5/atg7 independent sequestration mechanism. Alternatively the authors could perform experiments with p62-LIR-domain mutants.

3. The overall usefulness of PIMs as a useful probe for selective autophagy could be questioned, as only a small fraction of the cells actually shows PIM particles, 1% after 8 hours as I can understand. In addition, only a small percentage of these clusters actually convert from yellow to green.

4. If experimentally possible it would be useful to demonstrate whether these PIMs are actually ubiquitinated. These could be challenging due to the nature of the multimerization process leading to various protein weights, however, immunoprecipitation could be a possibility.

Minor points:

1. Autophagic sequestration of ubiquitinated aggregates using a tandem-GFP-cherry tag (on LC3 and p62) has been shown earlier and should be referenced.

2. It is not quite clear to me how the clusters observed in fixed cells were actually selected, as the authors state. Figure 2c shows a rather big cluster with high aggregate level my opinion. To clarify this matter, the authors could actually sort according to cluster size and get information on what cluster size might preferentially be sequestered.

Reviewer

Andreas Brech

Reviewer #2 (Remarks to the Author):

This manuscript introduces a new tool for the study of cellular mechanisms responsible for clearance of protein aggregates based on the addition of a chemical dimerizer to cells expressing a protein that contains multiple repeats of the FKBP* homodimerization motif along with GFP and Cherry. This strategy allows for the robust induction of protein aggregates that are subsequently ubiquitinated, recognized by autophagy adaptor proteins, sequestered into autophagosomes and delivered to lysosomes for degradation. The presence of GFP and Cherry proteins allows for aggregate visualization and assessment of lysosome delivering based on the loss of GFP fluorescence in the acid pH conditions of the lysosome lumen. The manuscript is well written and the tool that it describes should be useful for studying mechanisms that support clearance of protein aggregates. Such utility is furthermore established through analysis of cells lacking p62, an adaptor that recruits protein aggregates into autophagosomes.

I have just a few issues that should be addressed.

1. Please provide a reference to support the claim in the introduction that GFP is more sensitive to degradation than Cherry.

2. What is the basis for a loss of lysosomal GFP fluorescence that is detectable even in fixed cells where pH is no longer a factor?

3. Cells in which all PIMs converged to one single cluster were mentioned but were excluded from analysis. Approximately what fraction of cells exhibited this behavior?

4. Fig S3. The LAMTOR4 IF signal is weak. It is hard to evaluate the degree of triple co-localization between LAMTOR4, GFP and Cherry but there is clearly some when I would have expected none. Better detection of lysosomes would be helpful

Reviewer #3 (Remarks to the Author):

In this manuscript Janssen and co-workers report the chemically-induced formation of protein clusters named PIM (Particles Induced by multimerization). Formation of PIMs occur after cellular transient transfection of a construct encoding for mCherry and EGFP fused to several repeats of FKBP*, a domain with the capacity of homodimerizing upon treatment with the rapamycin-analog AP20187 (rapalog2). Fluorescence recovery after photobleaching (FRAP) experiments are used to show that stable multimerization does occur after rapalog2 treatment. The authors present the protein multimerization system as a useful tool to monitor and further study aggrephagy as the double tag strategy enables the system to monitor recombinant protein delivery to the lysosomal lumen. PIM cluster clearance is inhibited by Bafilomycin A1 treatment or knock-out of p62 in HeLa cells. PIM clusters colocalize with several endogenous selective autophagy markers (i.e. ubiquitin, p62 and LC3).

Tools to monitor selective autophagy are scarce. The development of new approaches is hence of interest to the field. Even though the chemically-induced protein multimerization approach has been previously published by the authors (as they report themselves) the modification of the original construct to mCherry-EGFP double-tagged is novel and potentially important. Even though the experiments shown in the manuscript are nicely presented, some relevant controls are lacking. The time points shown are not consistent throughout the manuscript, which makes difficult to compare the different figures. Image acquisition, presentation and analysis are problematic as the authors base their conclusions on aggregate clearance on the basis of the analysis of very few and highly selected protein clusters. Discarding unfavorable data leads to a biased interpretation of results and raises doubts about the use of the PIM system for a general study of aggrephagy.

The current manuscript is not of sufficient novelty or quality to warrant publication in Nature Communications. The authors should address the specific comments below to strengthen their data.

Specific comments:

Main text:

- The authors discuss that using aggregation-prone proteins to study aggrephagy might affect cellular health as well as the process of autophagy itself and propose their system as a good alternative. However, for this statement to be convincing authors should analyse general autophagy in their system in order to assess whether it is affected by the presence of PIM clusters.
- The introduction to the PIM construct is confusing. The original construct (from reference 10 and not really used in the paper) is presented in much more detail than the mCherry-EGFP double tagged version. The authors should refer to Fig.1a after the sentence "Next, we optimized the PIM construct for measuring autophagic flux by adding an EGFP domain (mCherry-EGFP-PIM)."
- Authors state that EGFP is more sensitive to degradation than mCherry without providing any data or reference.
- As a general concern, the use of transient transfection does not seem the best option for establishing a system that allows the characterization of selective autophagy (or any other system). Differences in transfection yield or expression levels between cells makes it very challenging to compare cells or experiments and to reach general conclusions. The generation and use of stable cell lines is highly recommended.
- A more thorough characterization of the PIM clusters would be appreciated. General approaches to study aggregates include filter-trap and differential detergent extraction assays.

- Author say: "Following their formation, PIM clusters began to accumulate in the perinuclear region (Fig. S2b)". Having the nuclear staining of the cells presented in such figure would reinforce this statement.

- In the text it can be read "After 8h already approximately $49\pm 3\%$ of the clusters was cleared, while after 16h an average clearance of $62\pm 3\%$ was reached (N=5 cells, Fig 1g)". It is not clear which are the clusters referred to here as the total number of clusters (Fig 1f) rather increase.

- "The amount of yellow clusters decreased from $103\pm 26\%$ to $39\pm 12\%$ in 15h [...]. During the first 12h, the total number of particles was reduced due to merging of PIMs". It seems that quantitation of number of particles is not a good indicator of the dynamics of the protein clusters. Alternative measurements (as number of clusters x cluster area, cluster-occupied cell area or cluster intensity average) could be more accurate and could provide a better description of the PIM behaviour if protein aggregates are fusing into aggregates-like bigger structures.

- Authors mention two possible scenarios for protein clusters: either PIMs merge into larger perinuclear clusters and thereafter parts of these are degraded or they undergo direct clearance of smaller clusters. It would be nice to further discuss these findings, to quantify the proportion of PIM that follow either way or to try and figure out the reasons behind the different degradation patterns. An estimated size ($\sim 3\mu\text{m}^2$) is given for large aggregates – the same kind of information should be provided for smaller aggregates. Scale bars are missing in Fig. S2d and S2e.

- The authors claim an efficient transfer of PIMs to lysosomes (an average of 4.4 events per hour per cell). It would be interesting to have the information of the total number of clusters to be able to estimate whether the provided average number actually corresponds to an efficient transfer of PIMs to the lysosomal lumen.

- 40-60 cells for conditions should not be considered as a "large number of cells" when studying cellular processes. Analysis of a higher number of cells would be desirable.

- When working with HeLa p62 KO cells, more than one clone should be analysed. In order to claim that clearance of PIM clusters is through selective autophagy the authors should show that starvation-induced autophagy is not altered in HeLa p62 KO cells (by showing LC3 blots or behaviour of double-tag LC3). Showing that PIM clearing is restored in p62 KO cells when p62 is rescued would be desirable.

- Even though the manuscript revolves exclusively around selective autophagic clearance of protein aggregates there is not a single reference to the work of Terje Johansen. At least the following paper should be mentioned: Bjørkøy G et al, J Cell Biol. 2005.

Figures:

Figure 1:

- Presenting the figures as the inverted contrast greyscale image of the mCherry channel does not allow getting the information that having a mCherry-EGFP double tag provides (as shown in Supplemental video 1). Even though a merged image of selected image is provided, the insert is too small to clearly appreciate details and still does not show the behaviour of PIM clusters in a whole cell.

- Fig 1c, images are lacking a scale bar.

- Fig 1d is showing the dynamics of red-only aggregates exclusively. It would be nice to know how frequent these are in the whole cell (compared to total number of clusters) and what is the dynamic of the total cluster population.

- Fig 1f and 1g: an N of 5 cells from 2 independent experiments does not seem enough to have reliable results. A N of at least 3 is desirable for all experiments. Also, a bigger number of cells should be analysed.

- Fig 1g, authors should clarify how the average fraction of red clusters is calculated (does it correspond to the number of red clusters divided by the number of total clusters? Is this average expressed "per cell"?).

Figure 2:

- It would be nice to see also example pictures for 1h time point.
- Concerns about normalization (see below).

Supplemental figure 1:

- Rapalog1 is not used in the manuscript. It is probably not needed in this figure.
- It would be nice to have a quantification of the bands and to know how many times the experiment has been performed.

Supplemental figure 2:

- Fig S2c, A N of at least 3 is desirable for all experiments
- Fig S2d and S2e, as mentioned for Fig. 1 presenting figures as the inverted contrast greyscale image of the mCherry channel precludes a more exhaustive analysis of the total clusters in the cell. Images are lacking a scale bar.
- Fig S2d, the way the left graph is presented is misleading. Take time 11h, whereas looking at the graph one sees that the total cluster number is decreasing and could assume that aggregates are being degraded by looking at the corresponding image one can see that what is happening instead is that the clusters are merging into bigger aggregate-like structures. As suggested before an alternative measure to "number of clusters" is recommended to represent the data.
- Fig S1e, whereas in the represented images one can see big differences in the sizes of clusters this information is not reflected in the right graph showing "particle size".

Supplemental figure 3:

- Some negative control of non-transfected cells and PIM transfected cells but non-treated with rapalog2 would be appreciated. By having such controls it would be possible to assess whether PIM overexpression itself alters the distribution of endogenous autophagic markers.

Supplemental figure 4:

- There is not reference to this figure in the main text.
- Information about how many times the experiment has been done is missing.

Supplemental video 2:

- Even though "time" is indicated in the video legend no time stamp is shown in the video.

Online methods:

- Treatment with rapalog2 is not comparable between live-cell imaging experiments (in which cells are kept in the presence of rapalog2 during the whole experiment) and fixed cells analysis (in which rapalog2 is washed off after 1h treatment). The authors should comment on the reasons behind this.

- Under "Online methods -Fluorescence microscopy" it says that "ten different positions were picked by searching in the red channel for cells that showed an average level of aggregates". It would be nice to know what does "average level" refer to, is it an average number, size, intensity? In any case, sample imaging should be performed in a blinded manner.

- During image analysis of live-cell experiments the authors only considered cells showing some yellow to red conversion and that could be followed for 16h, whereas cells showing late aggregation or PIMs convergence into a single cluster were discarded. For analysis of fixed cell images, cells with high aggregate levels resulting in no visible individual aggregates were discarded. This selection of clusters to be analysed will bias the results and conclusions raised from this study. It is for sure a way of getting nicer results and less variation, but it does not really describe the actual process happening with the total cluster population. Knowing the proportion corresponding to the selected analysed clusters (when compared to the total number of clusters present in a cell) would be interesting. As a general note, if the authors are proposing this system as a tool to trigger and monitor autophagy the system should be well characterized for the general cluster population rather than for specific selected clusters.

- "Values were normalized per cell by dividing all ratios by the average ratio of values >0.5 . This allows normalization by the average of the peak with the highest ratio which represents EGFP and mCherry positive clusters". To choose average ratio of values >0.5 to normalize data seems random and will create data dependence.

- In the immunofluorescence staining methods LAMP1 staining is mentioned, but that staining is never mentioned in the manuscript.

Response to the reviewer's comments

Reviewer #1:

Janssen and colleagues describe in their manuscript a new method for probing the turnover of inducible cytosolic aggregates by selective autophagy. They exploit the chemically-induced multimerization of mCherry containing 4 FKBP domains, which forms intracellular aggregates (PIMs, particles induced by multimerization) upon treatment with the rapalog AP20187 (rapalog2), in follow their turnover. In order to follow autophagic sequestration and lysosomal degradation they modify this construct with an additional GFP-tag to monitor quenching of GFP-fluorescence in autolysosomes/lysosomes.

Upon induction with rapalog2 they observe formation of yellow PIMs, which accumulate in the perinuclear region, and importantly after long term observation show a shift to the red color, indicating quenching of the GFP fluorophore. The dynamics of cluster formation and possible sequestration by the autophagic machinery was followed both by live cell imaging and on fixed cells. In addition, the authors perform quantification of live cell imaging data of clusters and follow their fate over a 16-hour period, showing that a certain percentage of these clusters are undergoing a shift from yellow to red fluorescence, indicating accumulation in an acidic environment. To further address the question whether some of the cluster actually are delivered to the autophagic pathway the authors perform colocalization experiment against autophagic adaptor proteins such as p62 and nbr1, as well as ubiquitin, a well established marker for autophagic targeting. They also label against the most widely used autophagic marker protein LC3 and the lysosomal marker Lamtor4. They conclude from these experiments that a certain proportion of the PIMs are actually transferred to lysosomal degradation by selective autophagy. Using a CRISPR-Cas-9 mediated knock-out of p62 they further show that this happens in a p62-dependent manner.

On a general basis I would like to comment that the manuscript could be of interest for the autophagic community, however that would depend on a more thorough investigation of the turnover these inducible cluster. There are several keys issues which would need to be addressed by the authors for a clear demonstration of autophagic turnover. There are some shortcomings on statistical significance of some experiments as noted below. With respect to reproducibility of the experiments I think it would be challenging for other labs to perform similar experiments, as many of the measurements are taken on a selection of induced clusters, however, the nature of this study somehow makes it mandatory to use certain selection criteria, especially when doing live cell imaging. So in my view this biased approach is necessary in this context. A greater impact on a broader audience of these data in the present form is in my opinion not given.

- We thank the reviewer for the careful assessment of the manuscript and for the constructive comments. In the revised manuscript, we have tried to address all these comments. Most importantly, we now demonstrate that aggregate clearance can also be analyzed using flow cytometry. This provides an unbiased alternative to analyze turnover and will allow other labs to readily reproduce our findings.

Mayor points:

1. The only clear evidence for delivery to the autophagic pathway is the actual colocalization with autophagic adaptor proteins (nbr1 and p62) and the rather poor localization data with LC3 and Lamtor4. The authors state actually that “only some” red clusters colocalize with both LC3 and Lamtor4, which is surprising with regard to their interpretation of autophagic turnover. These data lack also quantification, which should be straight forward to perform in these experiments, and would make a much stronger argument. On a technical note I was surprised to read that the fixation protocol for the IF images was only a 10 min PFA fixation, followed by overnight incubation with antibody in rather harsh conditions (Tween and BSA), which from my experience might be the reason for unsatisfactory labeling results.

➤ For the revised manuscript, we have quantified the colocalization of red and yellow clusters with different markers (i.e. ubiquitin, p62, NBR1, LC3, LAMTOR4, see new Figure 2). These results demonstrate recruitment of autophagy machinery markers and delivery to lysosomes. We agree that previous conditions could have been too harsh mainly for membrane localized proteins like LC3 and now performed stainings using a milder protocol for overnight incubation (3% BSA in PBS). In our opinion, it is not necessarily surprising that only a low percentage of aggregates are LC3 positive as the lifetime of the autophagosome with cargo might be short and LC3 is likely degraded in the lysosome. Additional support for activation of an autophagy pathway comes from the reduced clearance in p62 and Atg7 knockout cells (see below).

2. The authors should perform experiments with atg7 and atg5 siRNA-knock down to characterize the involvement of the core-autophagy machinery. Off course that would not exclude the possibility for an atg5/atg7 independent sequestration mechanism. Alternatively the authors could perform experiments with p62-LIR-domain mutants.

➤ Our original manuscript demonstrated reduced clearance in p62 knockout cells. We now also added data from Atg7 knockout cells (Supplemental Figure 6). These cells show a clear reduction, but not full abrogation, of clearance compared to control cells. This suggests that an Atg7 independent mechanism is possible, although less efficient.

3. The overall usefulness of PIMs as a useful probe for selective autophagy could be questioned, as only a small fraction of the cells actually shows PIM particles, 1% after 8 hours as I can understand. In addition, only a small percentage of these clusters actually convert from yellow to green.

➤ All cells showing detectable levels of transfection (based on fluorescence intensity) show PIMs after rapalog treatment. In addition, in many cells with indiscernible levels of fluorescence before rapalog addition aggregates also emerged. To clarify this, we have now added overview images in Supplemental Figure S2a. Live-cell imaging data indicated that 55% of these PIM-positive cells show clearance. Most importantly, we have now used flow cytometry to analyze clearance over the whole population in an unbiased manner (new Figure 4). These data demonstrate a clear shift towards lower GFP/mCherry ratios and demonstrate the value of our assay for screening applications.

4. If experimentally possible it would be useful to demonstrate whether these PIMs are actually ubiquitinated. These could be challenging due to the nature of the multimerization process leading to various protein weights, however, immunoprecipitation could be a possibility.

➤ We agree with the reviewer that a demonstration of direct ubiquitination would be useful. We therefore performed a His-ubiquitin pulldown to test whether our PIMs are directly being ubiquitinated. We indeed observed direct ubiquitination, but this was also observed without the addition of rapalog. Due to the inefficiency of the pulldown, we used HEK cells for these experiments. The extremely high overexpression in these cells caused premature aggregation of the construct without rapalog addition as observed by fluorescent microscopy. This precludes us from distinguishing whether ubiquitination occurs only when aggregates are formed or can also occur on the soluble protein. These data are now shown in Supplemental Figure 2.

Minor points:

1. Autophagic sequestration of ubiquitinated aggregates using a tandem-GFP-cherry tag (on LC3 and p62) has been shown earlier and should be referenced.

➤ We added references to tandem tagged LC3 and p62 in the introduction of the manuscript.

2. It is not quite clear to me how the clusters observed in fixed cells were actually selected, as the authors state. Figure 2c shows a rather big cluster with high aggregate level my opinion. To clarify this matter, the authors could actually sort according to cluster size and get information on what cluster size might preferentially be sequestered.

➤ We apologize if some of our methods were unclear. We now added overview images to show the overall diversity of aggregate expression (Supplemental Figure 2). In this figure, the asterisks mark cells that were excluded from analysis due to high aggregation levels. All other cells were analyzed and indeed display a variety of aggregation levels. In addition, the large cluster observed in Figure 2c is actually composed of multiple small clusters that can still be separated when scaling differently. Finally, inspired by the reviewer's suggestion, we now analyzed the size-dependent clearance and found that in control cells, there is no clear preference for any cluster size, because the size distribution of red clusters is similar to the size distribution of yellow clusters. Interestingly, this is not the case for P62KO lines where the red clusters appear to be smaller. This suggests that P62KO lines mainly have difficulty with clearing larger clusters.

Reviewer #2:

This manuscript introduces a new tool for the study of cellular mechanisms responsible for clearance of protein aggregates based on the addition of a chemical dimerizer to cells expressing a protein that contains multiple repeats of the FKBP homodimerization motif along with GFP and Cherry. This strategy allows for the robust induction of protein aggregates that are subsequently ubiquitinated, recognized by autophagy adaptor proteins, sequestered into autophagosomes and delivered to lysosomes for degradation. The presence of GFP and Cherry proteins in the allows for aggregate visualization and assessment of lysosome delivering based on the loss of GFP fluorescence in the acid pH conditions of the lysosome lumen. The manuscript is well written and the tool that it describes should be useful for studying mechanisms that support clearance of protein aggregates. Such utility is furthermore established through analysis of cells lacking p62, an adaptor that recruits protein aggregates into autophagosomes.*

I have just a few issues that should be addressed.

➤ We thank the reviewer for the careful and positive evaluation of our manuscript.

1. Please provide a reference to support the claim in the introduction that GFP is more sensitive to degradation than Cherry.

➤ We added the reference that supports the claim that GFP is more sensitive to degradation than mCherry (Katayama, H., Yamamoto, A., Mizushima, N., Yoshimori, T. & Miyawaki, A. GFP-like proteins stably accumulate in lysosomes. *Cell structure and function* 33, 1-12 (2008) and Kimura, S., Noda, T. & Yoshimori, T. Dissection of the autophagosome maturation process by a novel reporter protein, tandem fluorescent-tagged LC3. *Autophagy* 3, 452-460 (2007).).

2. What is the basis for a loss of lysosomal GFP fluorescence that is detectable even in fixed cells where pH is no longer a factor?

➤ Apart from the quenching of GFP fluorescence at low pH, GFP is also degraded in the lysosome by lysosomal proteases. mCherry, however, is much more resistant to lysosomal proteases and can therefore still be detected in the lysosomal lumen (see the reference above).

3. Cells in which all PIMs converged to one single cluster were mentioned but were excluded from analysis. Approximately what fraction of cells exhibited this behavior?

- Approximately 10% of the cells was excluded based on the fact that all clusters converged to one single point. Importantly, these cells were not excluded from the Flow Cytometry data that we added to the revised manuscript.

4. Fig S3. The LAMTOR4 IF signal is weak. It is hard to evaluate the degree of triple co-localization between LAMTOR4, GFP and Cherry but there is clearly some when I would have expected none. Better detection of lysosomes would be helpful.

- We have adapted our staining protocol and analyzed colocalization for both dual positive and Cherry only positive aggregates (new Figure 2). There are indeed some GFP and Cherry positive clusters that colocalize with LAMTOR4, which could indicate very recent fusion events. Importantly, the GFP intensity is often reduced relative to the mCherry fluorescence when compared to surrounding dual-color particles. In addition, because of the high density of lysosomes in the perinuclear region, we cannot exclude the occasional detection of false positives.

Reviewer #3 (Remarks to the Author):

In this manuscript Janssen and co-workers report the chemically-induced formation of protein clusters named PIM (Particles Induced by multimerization). Formation of PIMs occur after cellular transient transfection of a construct encoding for mCherry and EGFP fused to several repeats of FKBP, a domain with the capacity of homodimerizing upon treatment with the rapamycin-analog AP20187 (rapalog2). Fluorescence recovery after photobleaching (FRAP) experiments are used to show that stable multimerization does occur after rapalog2 treatment. The authors present the protein multimerization system as a useful tool to monitor and further study aggrephagy as the double tag strategy enables the system to monitor recombinant protein delivery to the lysosomal lumen. PIM cluster clearance is inhibited by Bafilomycin A1 treatment or knock-out of p62 in HeLa cells. PIM clusters colocalize with several endogenous selective autophagy markers (i.e. ubiquitin, p62 and LC3).*

Tools to monitor selective autophagy are scarce. The development of new approaches is hence of interest to the field. Even though the chemically-induced protein multimerization approach has been previously published by the authors (as they report themselves) the modification of the original construct to mCherry-EGFP double-tagged is novel and potentially important. Even though the experiments shown in the manuscript are nicely presented, some relevant controls are lacking. The time points shown are not consistent throughout the manuscript, which makes difficult to compare the different figures. Image acquisition, presentation and analysis are problematic as the authors base their conclusions on aggregate clearance on the basis of the analysis of very few and highly selected protein clusters. Discarding unfavorable data leads to a biased interpretation of results and raises doubts about the use of the PIM system for a general study of aggrephagy.

The current manuscript is not of sufficient novelty or quality to warrant publication in Nature Communications. The authors should address the specific comments below to strengthen their data.

- We thank the reviewer for the careful evaluation of our manuscript. To address the concerns of the reviewer, we have performed many additional (control) experiments. Most importantly, to address the concern that our procedures for image analysis could have biased our interpretation, we have now used flow cytometry to analyze clearance in a completely independent and much more unbiased set of experiments (Figure 4). The results demonstrate that many cells shift towards a lower GFP/mCherry ratio, consistent with delivery to lysosomes and subsequent degradation. In addition, P62KO cells showed reduced clearance. These new results demonstrate the value and robustness of our assay.

Specific comments:

Main text:

- *The authors discuss that using aggregation-prone proteins to study aggregophagy might affect cellular health as well as the process of autophagy itself and propose their system as a good alternative. However, for this statement to be convincing authors should analyse general autophagy in their system in order to assess whether it is affected by the presence of PIM clusters.*

➤ We did not intend to claim that cellular health or autophagy is not affected at all after formation of the PIM clusters. Nevertheless, overexpression of mutant htt or polyQ constructs can interfere specifically with the autophagic machinery and cause other loss-of-function phenotypes from the moment of overexpression. As this could hamper the study of the aggregophagy process, we propose our assay as an alternative method to trigger and follow the process. One advantage is that the moment of cluster formation can be timed and does not start from the moment of expression.

- *The introduction to the PIM construct is confusing. The original construct (from reference 10 and not really used in the paper) is presented in much more detail than the mCherry-EGFP double tagged version. The authors should refer to Fig.1a after the sentence “Next, we optimized the PIM construct for measuring autophagic flux by adding an EGFP domain (mCherry-EGFP-PIM).”*

➤ We have improved the introduction to our construct and added the reference to Figure 1a.

- *Authors state that EGFP is more sensitive to degradation than mCherry without providing any data or reference.*

➤ We added references that show that EGFP is more sensitive to lysosomal degradation than mCherry (Katayama, H., Yamamoto, A., Mizushima, N., Yoshimori, T. & Miyawaki, A. GFP-like proteins stably accumulate in lysosomes. *Cell structure and function* 33, 1-12 (2008) and Kimura, S., Noda, T. & Yoshimori, T. Dissection of the autophagosome maturation process by a novel reporter protein, tandem fluorescent-tagged LC3. *Autophagy* 3, 452-460 (2007).).

- *As a general concern, the use of transient transfection does not seem the best option for establishing a system that allows the characterization of selective autophagy (or any other system). Differences in transfection yield or expression levels between cells makes it very challenging to compare cells or experiments and to reach general conclusions. The generation and use of stable cell lines is highly recommended.*

➤ We agree with the reviewer that a stable cell line would be beneficial. We have made several attempts to generate stable cell lines but unfortunately these failed due to high homology between the FKBP domains. Although we made several sequence variants of all the domains, we so far did not succeed in generating stable cell lines. We also attempted several different approaches including viral delivery and the use of FlipIn lines. Importantly, in our revised manuscript we demonstrate the use of flow cytometry to analyze cluster clearance. These data demonstrate the robustness of our approach throughout different experimental replicates (Figure 4d), despite the use of transient transfection. Although we agree that the use of a stable cell line would be beneficial, we believe that still a lot could be gained and learned from using our current system.

- *A more thorough characterization of the PIM clusters would be appreciated. General approaches to study aggregates include filter-trap and differential detergent extraction assays.*

➤ We performed Filter-Trap analysis on our PIM clusters which showed PIM clusters are soluble in 2% SDS. After this we proceeded to use fractionation using different detergents. We found PIM clusters are still soluble in 1% SDS but not in 2% Triton. The results on the increased insolubility upon rapalog addition are shown in Supplemental Figure 2.

- Author say: “Following their formation, PIM clusters began to accumulate in the perinuclear region (Fig. S2b)”. Having the nuclear staining of the cells presented in such figure would reinforce this statement.

➤ We added the outline of the nucleus based on DAPI staining in Supplemental Figure 2b.

- In the text it can be read “After 8h already approximately $49\pm 3\%$ of the clusters was cleared, while after 16h an average clearance of $62\pm 3\%$ was reached ($N=5$ cells, Fig 1g)”. It is not clear which are the clusters referred to here as the total number of clusters (Fig 1f) rather increase.

➤ In the text we refer to the percentage of clusters that are red-only at that specific time point.

- “The amount of yellow clusters decreased from $103\pm 26\%$ to $39\pm 12\%$ in 15h [...]. During the first 12h, the total number of particles was reduced due to merging of PIMs”. It seems that quantitation of number of particles is not a good indicator of the dynamics of the protein clusters. Alternative measurements (as number of clusters \times cluster area, cluster-occupied cell area or cluster intensity average) could be more accurate and could provide a better description of the PIM behaviour if protein aggregates are fusing into aggresome-like bigger structures.

➤ Following the reviewer’s suggestion, we have now added alternative metrics to our manuscript. Supplemental Figure 3c shows the GFP/mCherry intensity ratio of the integrated intensity of all detected clusters. Furthermore, we added Flow Cytometry data in which the GFP and mCherry intensity of each cell is measured (Figure 4). This intensity-based method also demonstrates the decrease in GFP fluorescence compared to mCherry intensities over time and thus nicely complements the live-cell data analysis of the fraction of cleared clusters and the GFP/mCherry ratio over all clusters.

- Authors mention two possible scenarios for protein clusters: either PIMs merge into larger perinuclear clusters and thereafter parts of these are degraded or they undergo direct clearance of smaller clusters. It would be nice to further discuss these findings, to quantify the proportion of PIM that follow either way or to try and figure out the reasons behind the different degradation patterns. An estimated size ($\sim 3\mu\text{m}^2$) is given for large aggregates – the same kind of information should be provided for smaller aggregates. Scale bars are missing in Fig. S2d and S2e.

➤ We added a discussion of these findings in the text and quantified the proportion of cells that show each type of behavior. Most cells show clearance of small aggregates (70%) while 27% of cells first show clearance of small clusters followed later by clearance of larger clusters once they formed in the perinuclear area. Only 3% of cells show clearance of larger clusters only. These different degradation patterns are intriguing, but finding out the origin of these differences falls beyond the scope of this paper. The average size of aggregates is indicated in Supplemental Figures 2d and 2e. Scale bars were added.

- The authors claim an efficient transfer of PIMs to lysosomes (an average of 4.4 events per hour per cell). It would be interesting to have the information of the total number of clusters to be able to estimate whether the provided average number actually corresponds to an efficient transfer of PIMs to the lysosomal lumen.

➤ The total number of clusters over time is shown in Figure 1f. This panel also shows the number of red+green clusters and the number of red-only clusters.

- 40-60 cells for conditions should not been considered as a “large number of cells” when studying cellular processes. Analysis of a higher number of cells would be desirable.

➤ In the revised manuscript, we now also show flow cytometry data on 5000-10000 transfected cells per condition. We also adjusted our phrasing to no longer refer to 40-60 cells as a large number.

- When working with HeLa p62 KO cells, more than one clone should be analysed. In order to claim that clearance of PIM clusters is through selective autophagy the authors should show that starvation-induced autophagy is not

altered in HeLa p62 KO cells (by showing LC3 blots or behaviour of double-tag LC3). Showing that PIM clearing is restored in p62 KO cells when p62 is rescued would be desirable.

➤ We now added the analysis of a second P62KO clone, which was consistent with the results obtained using the first clone that we analyzed. Furthermore we added LC3 blots to show that starvation-induced autophagy is not affected in both HeLa P62 KO clones. Finally, we attempted to rescue loss of P62, but this resulted in the emergence of P62 clusters that sequestered our PIM construct even before rapalog addition. Therefore these experiments did not give a workable condition for analyzing rescue and were not further pursued.

- Even though the manuscript revolves exclusively around selective autophagic clearance of protein aggregates there is not a single reference to the work of Terje Johansen. At least the following paper should be mentioned: Bjørkøy G et al, J Cell Biol. 2005.

➤ We apologize for not including the work of Terje Johansen in our references. We added the reference to the suggested work and other relevant papers.

Figures:

Figure 1:

- Presenting the figures as the inverted contrast greyscale image of the mCherry channel does not allow getting the information that having a mCherry-EGFP double tag provides (as shown in Supplemental video 1). Even though a merged image of selected image is provided, the insert is too small to clearly appreciate details and still does not show the behaviour of PIM clusters in a whole cell.

➤ We now added the full merged images in Supplemental Figure 3.

- Fig 1c, images are lacking a scale bar.

➤ We added a scale bar in Figure 1c.

- Fig 1d is showing the dynamics of red-only aggregates exclusively. It would be nice to know how frequent these are in the whole cell (compared to total number of clusters) and what is the dynamic of the total cluster population.

➤ We are not entirely sure whether we understand this comment. Fig 1d shows the rapid decay of EGFP signal compared to the mCherry signal, indicative of entry into a lysosome.

- Fig 1f and 1g: an N of 5 cells from 2 independent experiments does not seem enough to have reliable results. A N of at least 3 is desirable for all experiments. Also, a bigger number of cells should be analysed.

➤ As requested by the reviewer, we now added data from a third experiment to our analysis.

- Fig 1g, authors should clarify how the average fraction of red clusters is calculated (does it correspond to the number of red clusters divided by the number of total clusters? Is this average expressed "per cell"?).

➤ The fraction of red clusters corresponds to the number of red clusters divided by the total number of clusters in a cell. The data was averaged per cell. We adapted the manuscript to make this more clear.

Figure 2:

- It would be nice to see also example pictures for 1h time point.

➤ In the revised manuscript, we added example pictures for the 1hr timepoint to the figure.

- Concerns about normalization (see below).

➤ See below.

Supplemental figure 1:

- *Rapalog1 is not used in the manuscript. It is probably not needed in this figure.*

➤ As rapalog1 is frequently used in many labs we decided to keep the data in the figure.

- *It would be nice to have a quantification of the bands and to know how many times the experiment has been performed.*

➤ We now added quantification of the bands based on 3 independent experiments.

Supplemental figure 2:

- *Fig S2c, A N of at least 3 is desirable for all experiments*

➤ We added analysis on a third independent experiment.

- *Fig S2d and S2e, as mentioned for Fig. 1 presenting figures as the inverted contrast greyscale image of the mCherry channel precludes a more exhaustive analysis of the total clusters in the cell. Images are lacking a scale bar.*

➤ We added a scale bar and the full color images to Supplemental Figure 3.

- *Fig S2d, the way the left graph is presented is misleading. Take time 11h, whereas looking at the graph one sees that the total cluster number is decreasing and could assume that aggregates are being degraded by looking at the corresponding image one can see that what is happening instead is that the clusters are merging into bigger aggresome-like structures. As suggested before an alternative measure to “number of clusters” is recommended to represent the data.*

➤ We added this example to illustrate that PIMs often cluster before clearance. This is why we do not utilize the decrease of total number of particles as an indication of clearance, but rather the increase of red clusters or the fraction of red clusters. Following the reviewer’s suggestion, we have now added alternative metrics to our manuscript. Supplemental Figure 3c shows the GFP/mCherry intensity ratio of the integrated intensity of all detected clusters. Furthermore, we added Flow Cytometry data in which the GFP and mCherry intensity of each cell is measured (Figure 4). This intensity-based method also demonstrates the decrease in GFP fluorescence compared to mCherry intensities over time and thus nicely complements the live-cell data analysis of the fraction of cleared clusters and the GFP/mCherry ratio over all clusters.

- *Fig S1e, whereas in the represented images one can see big differences in the sizes of clusters this information is not reflected in the right graph showing “particle size”.*

➤ The graphs shows the average particle size, although some clusters may increase in size due to merging this is not enough to greatly influence the average particle size. Furthermore the clusters that seem to merge into one big cluster in the 6 and 11hr timepoint in fig S1e can, when scaled differently, still be distinguished as independent clusters. We chose this scaling as details on the multiple smaller aggregates would otherwise be lost.

Supplemental figure 3:

- Some negative control of non-transfected cells and PIM transfected cells but non-treated with rapalog2 would be appreciated. By having such controls it would be possible to assess whether PIM overexpression itself alters the distribution of endogenous autophagic markers.

➤ In figure 2 non-transfected cells are now present in each of the pictures next to a cell showing clusters. Furthermore, we now added cells showing PIM overexpression without rapalog addition in Supplemental Figure 4.

Supplemental figure 4:

- There is not reference to this figure in the main text.

➤ We added a reference to this figure in the revised manuscript.

- Information about how many times the experiment has been done is missing.

➤ We added the requested information to the figure legend.

Supplemental video 2:

- Even though “time” is indicated in the video legend no time stamp is shown in the video.

➤ In the videos that we uploaded, time was indicated in the top left corner. When we download these videos, the time stamp is still there.

Online methods:

- Treatment with rapalog2 is not comparable between live-cell imaging experiments (in which cells are kept in the presence of rapalog2 during the whole experiment) and fixed cells analysis (in which rapalog2 is washed off after 1h treatment). The authors should comment on the reasons behind this.

➤ In almost all experiments, rapalog2 was washed off after 1hr of treatment. Only for the example given in Figure 1b rapalog was left on due to a lack of time for replacing the medium in between image acquisitions. We adapted this in our methods.

- Under “Online methods -Fluorescence microscopy” it says that “ten different positions were picked by searching in the red channel for cells that showed an average level of aggregates”. It would be nice to know what does “average level” refer to, is it an average number, size, intensity? In any case, sample imaging should be performed in a blinded manner.

➤ In Supplemental Figure 3 we now added on overview of cells expressing the PIM construct before and 1 and 8 hours after rapalog addition. All cells shown in these images are suitable for analysis, except the cells marked with an asterisk, which show very high aggregate levels. Importantly, we now added Flow Cytometry data to the revised manuscript, which validates our approach in an independent manner.

- During image analysis of live-cell experiments the authors only considered cells showing some yellow to red conversion and that could be followed for 16h, whereas cells showing late aggregation or PIMs convergence into a single cluster were discarded. For analysis of fixed cell images, cells with high aggregate levels resulting in no visible individual aggregates were discarded. This selection of clusters to be analysed will bias the results and conclusions raised from this study. It is for sure a way of getting nicer results and less variation, but it does not really describe the actual process happening with the total cluster population. Knowing the proportion corresponding to the selected analysed clusters (when compared to the total number of clusters present in a cell)

would be interesting. As a general note, if the authors are proposing this system as a tool to trigger and monitor aggrephagy the system should be well characterized for the general cluster population rather than for specific selected clusters.

➤ For the analysis of live-cell experiments, we indeed only analyzed cells that show clearance. We did not intend to suggest this represents the overall picture, but merely wanted to show that cells that show clearance are typically very efficient in clearing the clusters. This is useful information, when one sets out to study specific steps in the clearance process using live-cell imaging. Furthermore, when cells were analyzed, all clusters present in that cell were included in the analysis and not a proportion. We never selected specific clusters within cells for analysis, except for analyzing the speed of GFP loss in figure 1c-d. Finally, we now used Flow Cytometry to analyze the behavior of the general cell population in an unbiased manner. This revealed that a significant fraction of cells show efficient clearance, leading to an overall shift in GFP/mCherry ratio of cells (new Figure 4).

- *“Values were normalized per cell by dividing all ratios by the average ratio of values >0.5. This allows normalization by the average of the peak with the highest ratio which represents EGFP and mCherry positive clusters”. To choose average ratio of values >0.5 to normalize data seems random and will create data dependence.*

➤ We changed the data normalization method and now use Gaussian fitting to find the mean of the peak representing yellow clusters (i.e. most rightward peak). This mean was then used for data normalization. This new procedure did not change the outcome of the experiments.

- *In the immunofluorescence staining methods LAMP1 staining is mentioned, but that staining is never mentioned in the manuscript.*

➤ We removed this statement from the methods section.

Reviewers' comments:

Reviewer #1 (Remarks to the Author):

I have found the additional experiments and other improvements performed by the authors satisfactory and therefore recommend the revised manuscript for publication.

Reviewer #2 (Remarks to the Author):

I am satisfied with the responses to my concerns.

Reviewer #3 (Remarks to the Author):

This manuscript describes an assay that may be used to analyze aggrephagy (protein aggregate clearance by autophagy). The assay is based on transient transfection of two constructs encoding GFP and mCherry fused to an array of FKBP domains that multimerize upon addition of rapalog2 forming aggregate-prone PIMs.

In this revised manuscript Janssen and co-authors have done some additional experiments to strengthen their initial observations (e.g. the flow cytometry analysis in Figure 4 is nice). However, several of my initial concerns have not been addressed (as specified below) and in my opinion, the manuscript neither has sufficient novelty or quality to warrant publication in Nature Communications.

Specific comments:

As a transient transfection based approach neither gives very reproducible results or is very useful for most purposes (like HTP screening), the authors were asked to make (inducible) stable cell lines expressing the same constructs. They say they have tried different approaches to do so without success. This is of course acceptable, but makes their system less useful and thus even less interesting for the readers of Nature Communications.

The authors were also asked to provide more evidence for the aggrephagy specificity of their system. They show that p62 KO partially inhibits PIM clearance and were asked to i) show that this can be rescued by wild type but not a LIR mutant p62 and ii) that this is not due to an inhibition of general autophagy. This has not been done properly. i) The authors claim it is not possible to rescue as p62 forms aggregates that recruits the PIMs. This is likely the case if they overexpress p62 by transient transfection and they should therefore use e.g. a lenti-virus approach for these experiments. They should show that rescue is dependent of the LIR and UBA domain of p62. Point ii) has been addressed, but it has only been done once and must be quantified from several experiments.

I also asked the authors to provide evidence that the PIMs are indeed aggregated structures - this has been addressed using differential detergent fractionation, which is nice, but again only done once. It would have been nice to include treatment with BafA1 and blot for p62 as controls.

In general, there is an issue with proper controls throughout the manuscript. E.g. while most data shown is based on transient transfection in HeLa cells, they use ATG7 KO U2OS cells as controls for autophagy specificity. However, the PIM dynamics is different in HeLa and U2OS cells (as they mention on line 15 "although PIM clearance was generally slower in U2OS than in HeLa cells") and the reason for using U2OS cells for this experiment is not mentioned.

It is also problematic that some of the autophagy flux experiments (e.g. suppl Fig 6d) show data from different gels (bands cut), again with no quantification.

Another issue is the low number of cells (manual selection) used for quantification of PIMs by fluorescent microscopy. Moreover, this reviewer still think that presenting figures as inverted contrast of greyscale image of the mCherry channel is not the way to present the data. It is very difficult to check whether what the authors state is shown by the pictures (for instance number of red-only dots do not seem comparable at all between Fig. 2, Fig 3a and SFig 3a, all of them taken at 8h).

Minor issues:

In line 18 and line 65 the authors state "most clusters are delivered to the lysosome in a p62 and ATG7 dependent manner". The data presented do not seem enough to support that statement.

The authors show that rapalog2 treatment has no effect on mTOR activity, but should also show that basal autophagy is not influenced. They should also quantify the levels of p62, NBR1, Ub and LC3 in cells treated or not with rapalog2 to compare how much their pattern change.

In line 74 "EGFP domain" should be replaced by EGFP flurophore, protein...

In line 134 they say that LC3 only decorates a small subset of clusters "suggesting either rapid fusion between autophagosomes and lysosomes or the involvement of other ATG8 family proteins". This should be further checked with for instance BafA1 treatment or analysis with Ab against other members of the ATG8 family.

Western blots from Fig S5 and S6 should be quantified so that different conditions can be easily compared.

Authors see conversion to red in the absence of ATG7 (Fig. S6). As well as in the absence of p62 or the presence of BafA1 (Fig. 4). Such results make unclear the role of autophagy in the degradation of PIM clusters. Additional controls (i.e. ULK1 KD) should be perform in order to support the ideas the authors discuss could be behind this unexpected results.

Line 361. No mention of Hek293T cells in "cell culture and transfection" section.

In rebuttal: " The average size of aggregates is indicated in Supplemental Figures 2d and 2e." this is not the case.

Reviewer 3

Specific comments:

As a transient transfection based approach neither gives very reproducible results or is very useful for most purposes (like HTP screening), the authors were asked to make (inducible) stable cell lines expressing the same constructs. They say they have tried different approaches to do so without success. This is of course acceptable, but makes their system less useful and thus even less interesting for the readers of Nature Communications.

- We respectfully disagree with this assessment and feel that, in particular, our new FACS data shows the robustness of our approach. Following the editorial guidance, we did not further attempt generating stable cell lines.

The authors were also asked to provide more evidence for the aggrephagy specificity of their system. They show that p62 KO partially inhibits PIM clearance and were asked to i) show that this can be rescued by wild type but not a LIR mutant p62 and ii) that this is not due to an inhibition of general autophagy. This has not been done properly. i) The authors claim it is not possible do rescue as p62 forms aggregates that recruits the PIMs. This is likely the case if they overexpress p62 by transient transfection and they should therefore use e.g. a lenti-virus approach for these experiments. They should show that rescue is dependent of the LIR and UBA domain of p62. Point ii) has been addressed, but it has only been done once and must be quantified from several experiments.

- Following the editorial guidance, we did not pursue lenti-virus approaches to achieve rescue of p62 knockdown. For the first revision, we already performed the LC3 flux experiment in HeLa p62KO cells in triplicate. We now added the quantification of these western blots (Supplemental Figure 5d-e). The results show that autophagic flux is not impaired in p62KO cells.

I also asked the authors to provide evidence that the PIMs are indeed aggregated structures - this has been addressed using differential detergent fractionation, which is nice, but again only done once. It would have been nice to include treatment with BafA1 and blot for p62 as controls.

- In our first revised manuscript, we stated that Triton fractionation was performed three times but the SDS fractionation only once. We now have only repeated the fractionation in SDS three times and added quantification (Supplemental Figure 2e-f). The results did not change. Following editorial guidance, we did not include BafA1 treatment and p62 blotting to this set of experiments.

In general, there is an issue with proper controls throughout the manuscript. E.g. while most data shown is based on transient transfection in HeLa cells, they use ATG7 KO U2OS cells as controls for autophagy specificity. However, the PIM dynamics is different in HeLa and U2OS cells (as they mention on line 15 "although PIM clearance was generally slower in U2OS than in HeLa cells") and the reason for using U2OS cells for this experiment is not mentioned.

- We chose to use U2OS cells, as these cell lines were already available in the lab. Generation of KO cell lines would not have been possible in the given time frame for initial revisions, considering the considerable amount of additional experiments that were requested.

It is also problematic that some of the autophagy flux experiments (e.g. suppl Fig 6d) show data from different gels (bands cut), again with no quantification.

- We repeated these experiments to make sure the data was on one gel and also added quantification (Supplemental Figure 6d-e).

Another issue is the low number of cells (manual selection) used for quantification of PIMs by fluorescent microscopy.

- For Figure 3, we analyzed 40-60 cells per conditions, which is not necessarily a low number of cells for quantification. To circumvent the concerns about manual selection and low cell numbers, we added FACS and show very similar results for a large number of cells (20,000 – 50,000 per condition). Following the editorial guidance, we did not further increase the number of cells per experiment.

Moreover, this reviewer still think that presenting figures as inverted contrast of greyscale image of the mCherry channel is not the way to present the data. It is very difficult to check whether what the authors state is shown by the pictures (for instance number of red-only dots do not seem comparable at all between Fig. 2, Fig 3a and SFig 3a, all of them taken at 8h).

- The only time we choose to present only a greyscale image is in Fig. 1a. The full merges of Fig. 1a are available in supplemental Fig. S3a as requested by the reviewer. Furthermore, there is some inherent variability between cells that we do not wish to hide. The example in Fig. 3a shows 29% clearance, while the average is 21%. We chose this example because smaller red aggregates are usually hardly visible in a merged image. This is also the reason we chose for inverted contrast in Fig. 1a. For the example stainings in Fig 2 other criteria were used like nice spread cell, good staining, and the presence of a neighboring cell without aggregates. Following the editorial guidance, we did not make additional changes to the presentation of our figures.

Minor issues:

In line 18 and line 65 the authors state "most clusters are delivered to the lysosome in a p62 and ATG7 dependent manner". The data presented do not seem enough to support that statement.

- We have rephrased these statements.

The authors show that rapalog2 treatment has no effect on mTOR activity, but should also show that basal autophagy is not influenced. They should also quantify the levels of p62, NBR1, Ub and LC3 in cells treated or not with rapalog2 to compare how much their pattern change.

- Following the editorial guidance, we have concentrated on testing LC3 flux in the absence and presence of rapalog2 and found no difference between these two conditions (Fig. S1b). In addition, we tested the levels of p62 and ubiquitinated proteins and observed no changes upon treatment with rapalog2 (Fig. S1c,d).

In line 74 "EGFP domain" should be replaced by EGFP flurophore, protein...

- As requested by the reviewer, we changed EGFP domain to EGFP fluorophore in line 74.

In line 134 they say that LC3 only decorates a small subset of clusters "suggesting either rapid fusion between autophagosomes and lysosomes or the involvement of other ATG8 family proteins". This should be further checked with for instance BafA1 treatment or analysis with Ab against other members of the ATG8 family.

- For the rerevised manuscript, we now also quantified LC3 colocalization with PIMs in the presence of BafA1 and found that the overlap was increased to about 50%, indeed suggesting that autophagosomes are short lived (Fig. 2e).

Western blots from Fig S5 and S6 should be quantified so that different conditions can be easily compared.

- We added quantifications of these western blots.

Authors see conversion to red in the absence of ATG7 (Fig. S6). As well as in the absence of p62 or the presence of BafA1 (Fig. 4). Such results make unclear the role of autophagy in the degradation of PIM clusters. Additional controls (i.e. ULK1 KD) should be perform in order to support the ideas the authors discuss could be behind this unexpected results.

- In our opinion these results are not necessarily unexpected. P62 is not the only autophagy receptor and other receptors, e.g. NBR1, could also be involved. Furthermore, recent work has demonstrated that Atg7 is not essential for autophagosome formation (Nishida et al., Nature 2009, Tsuboyama et al., Science 2016). We agree that the shift in BafA1 treated cells in the FACS results is somewhat unexpected. This could be caused by partial quenching of EGFP when clustered or by FRET between EGFP and mCherry. This is why we set the threshold for being considered cleared quite high (Fig. 4a-e). We believe we have convincingly demonstrated that most aggregates are cleared through autophagy, because individual aggregates rapidly lose EGFP fluorescence and red-only aggregates colocalize with lysosomes, while both events are significantly reduced upon knockdown of p62 or treatment with BafA1. Given the verdict of the other two experts and following the editorial guidance, we did not initiate additional experiments to further demonstrate that most aggregates are cleared by autophagy.

Line 361. No mention of Hek293T cells in "cell culture and transfection" section.

- We have added HEK293T cells to this section.

In rebuttal: " The average size of aggregates is indicated in Supplemental Figures 2d and 2e." this is not the case.

- We apologize for this error in figure referencing. We intended to refer to Supplemental Figure 3d and 3e.